# Ionic Liquids in Pharmaceutical and Biomedical Applications: A Review

**DOI:** 10.3390/pharmaceutics16010151

**Published:** 2024-01-22

**Authors:** Yue Zhuo, He-Li Cheng, Yong-Gang Zhao, Hai-Rong Cui

**Affiliations:** 1School of Biomedical Science and Engineering, South China University of Technology, Guangzhou 511442, China; zhuoyue1102@gmail.com; 2Shanghai Municipal Center for Disease Control & Prevention, Shanghai 200336, China; chengheli2017@126.com; 3College of Biological and Environmental Engineering, Zhejiang Shuren University, Hangzhou 310015, China; 4College of Life Sciences, Wuchang University of Technology, Wuhan 430223, China

**Keywords:** ionic liquids, synthesis, drug solubility, drug analysis, drug delivery, antimicrobial effect

## Abstract

The unique properties of ionic liquids (ILs), such as structural tunability, good solubility, chemical/thermal stability, favorable biocompatibility, and simplicity of preparation, have led to a wide range of applications in the pharmaceutical and biomedical fields. ILs can not only speed up the chemical reaction process, improve the yield, and reduce environmental pollution but also improve many problems in the field of medicine, such as the poor drug solubility, product crystal instability, poor biological activity, and low drug delivery efficiency. This paper presents a systematic and concise analysis of the recent advancements and further applications of ILs in the pharmaceutical field from the aspects of drug synthesis, drug analysis, drug solubilization, and drug crystal engineering. Additionally, it explores the biomedical field, covering aspects such as drug carriers, stabilization of proteins, antimicrobials, and bioactive ionic liquids.

## 1. Introduction

Ionic liquids (ILs), which are also called molten salts, are a class of bulky and asymmetric organic cations and organic or inorganic anionic compounds with melting points generally below 100 °C [1,2,3]. Compared with traditional organic solvents, ionic liquids have many unique properties, and their physical and chemical properties can be regulated by adjusting cations or anions [4,5]. Different combinations of cations and anions can form a huge number of ILs, and there are various ways to categorize them. According to the different ionic structure, traditional ILs are mainly divided into substituting imidazole, pyridine, pyrrolidine, quaternary ammonium salt, pyrrolidinium, quinolinium, etc. Commonly used anions are tetrafluoroborate, hexafluorophosphate, chloride, bromine, nitrate, acetic acid, methyl sulfate, and so on (Figure 1) [6,7,8].

ILs with different chemical structures and properties can be briefly divided into three generations according to the order and age of their discovery. The research on ionic liquids has undergone development from the first generation to the third generation. The first generation, mainly used in the electroplating field, essentially combined dialkylimidazolium and alkylpyridinium cations with metal halide anions. These ILs with special physical properties, such as high thermal stability, low melting point, and broad liquidity, can be used to prepare functional solvents instead of certain organic solvents. Most first-generation ILs suffer from low biodegradability, high toxicity to the aquatic environment, and high preparation costs [9,10]. The second generation is stable in water and air, and it is synthesized from cations (e.g., dialkylimidazolium, alkylpyridinium, ammonium, and phosphonium) and anions (e.g., tetrafluoroborate and hexafluorophosphate). These ILs with unique chemical properties can be used to prepare functional materials. By adjusting and modifying anions and cations and their substituents, physical and chemical properties such as melting point, viscosity, thermal stability, hydrophilicity, solubility, toxicity, and biodegradability can be customized [11,12]. The third generation of ILs employs some natural sources of anions (such as amino acids, fatty acids, etc.) and cations (such as choline). In addition to good physical and chemical properties, the third generation of ILs also has low toxicity and good biodegradability. With the emergence of the third generation of ILs, the research on the application of ILs in bio-medicine is increasing gradually [11,13,14].

ILs have the advantages of designability, green non-toxicity, high stability, high solubility, and specific biological activities. The number of articles involving ILs is impressive and is growing significantly, and some of them show that ionic liquid can promote the reaction process, increase the yield, and reduce environmental pollution [2,11,15]. They have been successfully used in green solvents, drug delivery, and drug synthesis, and other fields show great application prospects [16,17]. Furthermore, their high tunability and good solubilization provide a new strategy for addressing the problems of poor solubility, unstable crystal form, poor biological activity, and low drug delivery efficiency in the pharmaceutical field. It is interesting to note that the focus of many IL studies now evolves in the direction of life sciences and medicine, leading to the emergence of bio-medical applications as one of the major research trends in ILs (Figure 2) [17,18,19,20,21,22].

## 2. Ionic Liquids in Pharmaceutical Applications

In recent years, the application of different types of ionic liquids in the pharmaceutical field on an industrial basis has received a great deal of attention worldwide, aiming to address the polymorphism, limited solubility, poor permeability, instability, and low bioavailability of crystalline drugs.

### 2.1. Application of Ionic Liquids in Drug Synthesis

#### 2.1.1. Solvents and Catalysts in Drug Synthesis

Since the early 1990s, domestic and foreign researchers have generated great research interest in green solvents, coinciding with the rise of green chemistry [23]. Green solvents are mainly water, ionic liquids, low eutectic solvents, supercritical CO_2_. From an environmental point of view, the use of large amounts of volatile and toxic organic solvents in the production process of drugs is an issue that deserves serious consideration. Recently, it is attracting an upsurge of interest in the potential use of ILs as replacements of conventional volatile organic solvents for drug synthesis under mild conditions [24,25]. ILs are used in numerous chemical processes, including their fascinating application in the synthesis of pharmaceutical substances and drugs. They offer many advantages, such as good thermal and chemical stability, strong solubility, polarity, and hydrophilicity. In order to meet specific needs, ions can be functionalized by introducing the active groups, such as hydroxyl and ether groups, to form functionalized ILs with low toxicity and easy degradation [18,26,27]. Compared with common organic solvents, ILs are characterized by low steam pressure and weak volatility, which can meet the needs of general chemical reactions and separation and purification operations. When ILs are used as solvents in drug synthesis, ionic liquid-organic systems are formed. Firstly, volatile components, such as organic solvents in the reaction, are removed through vacuum distillation using the low vapor pressure of the ILs in the purification procedure. There are two main categories of ILs: hydrophilic ILs and hydrophobic ILs. For hydrophilic ILs, the use of organic solvents to extract the reaction products enables the separation of other components of the ILs system and the recycling of the ILs. Generally, hydrophobic ILs are insoluble in certain organic solvents, such as ether, hexane, etc. Two phases will be formed when some organic solvents that are not mutually soluble with ILs are added to the ILs solution, and those hydrophobic products that are difficult to volatilize, will be extracted into the organic solvent, so as to achieve the purpose of separation and recovery of ILs. At the same time, ILs are suitable for reaction systems that require high vacuum or high temperature due to their properties of good stability, large solubility, ease of recycling, and resistance to burning and exploding. ILs contribute to reaching chemical equilibrium more rapidly, increasing the reaction rate, reducing reaction steps, and increasing product yields, selectivity, and efficiency of the process compared to conventional solvents. They are an ideal medium for reactions using transition metal catalysts with high polarity and weak bonding ability. Recently, ILs have been used as green solvents, reagents, catalysts, and enantioselectivity enhancers in the synthesis of various active pharmaceutical ingredients [28,29]. Marcin et al. have examined the efficiency of employing ILs with various cations (ammonium-, piperidinium-, pyridinium-, and imidazolium-based), each having a distinct structure of alkyl side-chains, pairing with the [N(Tf)_2_] anion as novel co-solvents for the reaction of acetic anhydride with curcumin under moderate conditions. This study investigated the potential application of bis(trifluoromethylsulfonyl)-imide-based ionic liquids as recyclable reaction media for the esterification of curcumin, obtaining curcumin diacetate without the use of additional solvents. The results demonstrate that [N(Tf)_2_]-based ILs are good solvents for the curcumin esterification reaction, and the presence of ionic liquids induces a significant change in the rate of the reaction, reducing the esterification time to only 15 min under optimized conditions. Additionally, the highest yield of curcumin diacetate (98%) was obtained in conditions containing [C_4_C_1_im][N(Tf)_2_], and [C_6_C_1_im][N(Tf)_2_] had the ability to be recycled three times, with no significant changes in catalytic activity. At the end of the reaction, depressurized filtration was performed to separate the ionic liquid from the reaction mixture. The filtrate containing ionic liquid was washed three times with ethyl acetate and then dried in a vacuum dryer for 24 h at 50 °C in preparation for the second cycle [30]. Sangwan et al. synthesized 1,8-dioxooctahydroxanthene derivative (3a–3k) using ILs 1-butyl-3-methylimidazoliumtetrafluoroborate (BMIF), 1-butyl-3-methylimidazoliumbromide (BMIB), and 1-butyl-3-methylimidazoliumchloride (BMIC) without a solvent. The ecofriendly and efficient method has an excellent yield of up to 90%. Then, the in vitro antitumor activity of all synthesized compounds against human lung cancer cell lines(A549) was determined, finding that 9-(3,4-dimethoxyphenyl)-3,3,6,6-tetramethyl-3,4,5,6,7,9-hexahydro-1H-xanthene-1,8(2H)-dione showed a marked inhibitory effect on the tested cancer cell line A549 [31].

The industrial synthesis of pharmaceutical compounds usually involves catalysts, the catalytic effect of which determines the yield of the final product but also causes organic pollution of the final product. In addition to being reaction solvents, ILs also have certain catalytic and selective effects on some reactions. They are widely used in many drug synthesis processes, such as heterocyclic synthesis, alkylation, oxidation, and dehydration, due to their excellent catalytic properties, environmental friendliness, economy, short reaction time, recyclable catalyst, simple operation, safe and mild reaction conditions, etc. [32,33,34]. Bhongale et al. prepared 1,3-disulfonic acid imidazolium hydrogen sulfate acting as a catalyst aiming at the O-alkylation of hydroquinone (HQ) into 4-methoxyphenol, which is a key intermediate in many pharmacological applications, under mild reaction conditions, reflux temperature, and atmospheric pressure. This method possesses the advantage of high substrate conversion, with a 93.79% yield for the desired product and high product selectivity. After the completion of the reaction, the remaining methanol was completely recovered through vacuum distillation. An excess of dichloromethane was then added to the residue to dissolve all organic matter, followed by centrifugation. The washing procedure was repeated once. The obtained catalyst residue was heated to 373 K for 120 min under vacuum and then reused. The catalyst exhibits excellent reaction rates, selective single-product formation, ease of separation, and recyclability, with effective recyclability through the fifth cycle without a significant loss in its activity [35]. Diarylmethanes are important architectures that are widely present in biologically and physiologically active compounds. Wu et al. reported a hydrogen bonding and acid co-catalytic strategy, which is a highly efficient and green catalytic system, for the benzylation of arenes with benzyl alcohols using Brønsted acidic ILs. It was found that ILs, such as 1-propylsulfonic acid-3-methylimidazolium trifluoromethanesulfonate ([SO_3_H-PMIm][OTf]), could catalyze this reaction efficiently and exhibit better performance than acidic catalysts (e.g., triflic acids, H_2_SO_4_), providing a range of diarylmethanes in good to excellent yields. It was found that IL cations catalytically activate benzyl alcohol through acid and/or hydrogen bonding to form benzyl cations, and anions act as hydrogen bonding acceptors to activate C-H bonds on the benzene rings of aromatic hydrocarbons, synergistically realizing the benzylation of aromatic hydrocarbons to obtain diarylmethane. Furthermore, the reaction occurs at the interface between the IL-based phase and the aromatic-based phase, and the IL separates spontaneously after the reaction due to the unique phase behavior of the reaction system [36].

#### 2.1.2. Reaction Mechanism Analysis of Ionic Liquids in Drug Synthesis

Imidazolium cation can react with Pd complexes to form carbene-Pd, which in some cases serves as a good catalyst for Heck or Suzuki reactions [37]. Mo et al. found that Heck arylation of electron-rich olefins could be accomplished with a variety of aryl bromides and iodides in excellent regioselectivity using the ionic liquid 1-butyl-3-methylimidazolium tetrafluoroborate ([bmim][BF4]) as a solvent without the need for aryl triflates or halide scavengers. Since ILs are composed entirely of ions, electrostatic interaction is more likely to produce a pd-olefin cation and a halide anion from two neutral precursors than a neutral pd-olefin intermediate from the same precursor. They concluded that the use of ILs as solvents could facilitate the ionic pathway by generating branched olefins without the need for halide scavengers. Moreover, the acidic C2-H proton of the imidazolium ring forms hydrogen bonds with halide anions, which may also contribute to the acceleration of the ionic reaction by promoting the dissociation of halide anions from palladium and stabilizing it [38].

As we all know, organic esters are very important pharmaceutical intermediates. Protophilic amide ionic liquid (PAIL) had already been used for esterification, and it can stabilize the intermediates in the esterification process. Xu et al. investigated the esterification reaction assisted by PAIL and its catalytic mechanism. The key two steps in the esterification reaction are the protonation of the acid and the nucleophilic attack of the protonated acid by the alcohol. PAIL accelerated the esterification reaction by stabilizing the protonated acetic acid by sharing its electron cloud with protonated carbonyl and limiting the alcohol protonation. The addition of inorganic acids, the conventional catalyst, increased the chemical shift of carbon on the carbonyl of acetic acid, whereas PAIL did not have this effect. In [DMF]^+^HSO_4_^−^, acetic acid was not protonated by H+, but rather attacked by [DMF]^+^, forming new stable intermediates. The carbonyl of DMF shares electrons with the protonated carbonyl group of acetic acid, giving it a higher electron density than that of the protonated carbonyl group in inorganic acids. In addition, PAIL is less able to protonate alcohols compared to inorganic acids. Therefore, the alcohols in PAIL attack the protonated carbonyl group of acetic acid more readily than the alcohols in inorganic acids. Based on these two aspects, PAIL accelerates the esterification reaction [39].

ILs can promote the formation of active reaction substrates during reactions. Akbari et al. synthesized a-aminophosphonates from aldehydes and ketones using a sulfonic acid functionalized IL as a Brønsted acid catalyst. The reaction mechanism is the formation of an active imine induced by IL. The subsequent reaction of the active amine with the added phosphite yields a phosphonium intermediate. Finally, the phosphonium intermediate reacts with water generated during imine formation to yield an aminophosphonate and methanol. ILs also contribute to some hydrogenation reactions [40]. Nian et al. studied the catalytic hydrogenation of citronellal to menthol using Cu/ZrO_2_-SiO_2_ as a catalyst in IL. The results showed that the cations in the ILs formed hydrogen bonds with the carbonyl group in the citronellal molecule, which made it easier to isomerize citronellal to menthol. In particular, the acidity-adjustable [bmim][AlmCln] ILs effectively provided the Lewis acid conditions for the isomerization of citronellal, promoting the conversion of citronellal towards the production of menthol in competitive hydrogenation [41].

The Knoevenagel condensation is one of the most useful carbon–carbon bond-forming reactions in organic syntheses. Xin et al. investigated the Knoevenagel condensation reaction of various aromatic aldehydes with active methylene compounds using a synthesized cyclic guanidinium lactate ionic liquid as the medium at room temperature, resulting in a high yield of over 90% in 1–7 min. It was found that the substitution group of the aromatic aldehyde, whether electron-donating or electron-withdrawing,, had little effect on the reaction. Additionally, the activity of methylene was the key factor influencing reaction speed [42]. Saha et al. synthesized 2-aryl benzimidazoles through the condensation reaction of o-phenylendiamine and several substituted aromatic aldehydes, promoted by the ionic liquid [pmim]BF4. The nature and position of substituents on the aryl ring were found to have little effect on reactivity. The IL acts as both a catalyst and a reaction medium in the reaction. Firstly, [pmim]BF4 activates an aldehyde and exposes it to the nucleophilic attack of o-phenylenediamine, forming a monoaldimine. Subsequently, cyclization of the monoaldehyde imine is followed by oxidative dehydrogenation in the air to generate benzimidazole [43].

### 2.2. Application of Ionic Liquids in Drug Extraction and Analysis

In recent years, ILs have attracted much attention because of their unique advantages in the separation of drugs due to their tunable structure [44,45]. Natural products are a class of secondary metabolites of natural organisms from a wide range of sources, including animals, plants, marine organisms, microorganisms, etc. They are very important for human health, greatly promoting the development of human medicine and health. In recent years, there has been a growing demand for therapeutic drugs derived from natural products. Natural products are extracted traditionally through decoction, reflux, maceration, and Soxhlet extraction methods by using mainly volatile organic solvents such as petroleum ether, ethyl acetate, and acetone, which have the disadvantages of low extraction efficiency, environmental pollution, non-renewability, and the production of easily generated toxic by-products. Novel renewable ionic liquids and their aqueous solutions, a class of safe and powerful media, have emerged as promising solvents to overcome some of the concerns associated with the use of volatile organic compounds in the field of extraction and separation. Some of the findings bring new perspectives on the use of appropriate aqueous solutions of ILs in place of commonly used volatile organic solvents for the extraction of compounds with biological activity from natural products, without the need for additional product recovery/separation steps [45]. Natural 7-hydroxymatairesinol (HMR) is a mixture of two stereochemical different C-7 isomers, namely (7R,8R,8′R)-(-)-allo-hydroxymatairesinol (HMR_1_) and (7S,8R,8′R)-(−)-7-hydroxymatairesinol (HMR_2_). HMR and its derivatives have been extensively investigated for pharmaceutical applications due to their anticarcinogenic and antioxidative properties. Ferreira et al. studied aqueous solutions of analogues of glycine-betaine ionic liquids (AGB-ILs) as alternative solvents for the extraction of HMR from knots of Norway spruce trees, instead of volatile organic solvents (e.g., acetone, water–acetone, and water–ethanol mixtures). The extraction operational parameters were optimized using response surface methodology (RSM). The extraction rate of AGB-ILs aqueous solution was as high as 9.46 wt% at 1.5 M and 25 °C, which is higher than that obtained with volatile organic solvents at higher temperatures and with longer extraction times. In addition, they evaluated the cytotoxicity of aqueous IL solutions containing HMR extracts in macrophage cell lines, as well as their anti-inflammatory potential by reducing lipopolysaccharide-induced oxidative stress in cells. The images (Figure 3C) obtained through fluorescence microscopy showed the effect of the precipitated HMR-rich extract, [(C_2_)_3_NC_2_] Br, and the HMR extract in an aqueous solution, [(C_2_)_3_NC_2_]Br (all at 100 µg mL^−1^), on cellular oxidative stress. Cell nuclei stained with Hoechst 33342 were blue, while cellular reactive oxygen species (ROS) would appear green through the H2DCFDA probe. The lack of green coloration in the second and third lines of Figure 3C shows no oxidative damage in the cells of all samples at 100 µg mL^−1^. Figure 3D shows the capacity of the precipitated HMR and that of the HMR in IL aqueous solutions to reduce lipopolysaccharide (LPS)-induced oxidative stress. The decrease of intensity in the cells’ green color shows that HMR has indeed considerable cellular antioxidant activity, reducing the LPS-induced production of ROS. The studies on the cytotoxicity of the [(C_2_)_3_NC_2_]Br aqueous solutions support their biocompatible nature and the high antioxidant activity of the HMR rich extracts. The antioxidant and anti-inflammatory potential of HMR-rich aqueous IL solutions was found to be more promising than that of the recovered HMR-rich solid extracts [46].

Laboratory analysis typically requires multiple steps, including extraction, purification, and pre-concentration, to ensure the reliable determination of analytes in the sample. Sample preparation methods are usually time-consuming, representing the limiting step of the overall analytical process in most cases. Efficient and effective sample preparation techniques are critical in analytical chemistry. Dispersive solid phase extraction has been widely used in the detection of trace drugs because of its simple operation, high efficiency, fast speed, and excellent kinetic properties. Hassan et al. synthesized two magnetic ionic liquids, (Z)-octadec-9-en-1-aminium tetrachloroferrate (III) and (Z)-octadec-9-en-1-aminium trichlorocobaltate (II), and used them as adsorbents for dispersive micro-solid phase extraction, coupled with HPLC-DAD for the separation, preconcentration, and quantification of the carbamazepine drug in urine and environmental water samples at room temperature. During the extraction process, the sorbent forms micelles in water with a hydrophobic core and hydrophilic perimeter. The hydrophilic perimeter of the magnetic ionic liquid adsorbent extracts the carbamazepine from the sample matrix by inducing a dipole moment to interact with its aromatic ring. Then, 500 µL of acetonitrile was used for the desorption of CBZ from the sorbent, and the desorbed analytes in the acetonitrile solvent were injected into the HPLC-UV for analysis. The magnetic ILs adsorbent can be reused at least five times and consumes a small amount of organic solvent. The established detection method using the prepared ionic liquid as extractant was simple, sensitive, efficient, and highly selective [47].

The analysis of a large number of complex and diverse compounds has always been a hot and difficult issue in the research of natural products due to their high biological diversity and chemical structure complexity [48]. Additives, such as formic acid, acetic acid, phosphoric acid, phosphate, alkylamine, alkyl sulfonate, triethylamine, or ammonium acetate, are generally added to the mobile phase to obtain a better peak shape and separation effect during liquid chromatography analysis, which will bring irreversible damage to the chromatographic column at the same time. Therefore, the selection of suitable mobile phase additives during chromatographic analysis is an urgent problem. In this context, ionic liquids have received a lot of attention and have been widely used as additives to improve target separations [49,50,51]. Ding et al. developed a reversed-phase high-performance liquid chromatographic method for the simultaneous determination of jatrorrhizine, palmatine, and berberine in Phellodendron bark using 1-hexyl-3-methylimidazolium trifluoroborate as a mobile phase additive. The effects of the concentration of 1-hexyl-3-methylimidazolium trifluoroborate and the pH of the mobile phase on the chromatographic behaviors of the three compounds were investigated. The optimal mobile phase was acetonitrile–water (25:75, *v*/*v*) containing 16 mmol L^−1^ 1-hexyl-3-methyl-imidazole-trifluoroborate at pH 3.0. Furthermore, the possible mechanism was also explored and discussed to explain the role of ionic liquids as the mobile phase additives. When ILs were used as mobile phase additives, they could effectively shield the residual silanols of the stationary phase and improve the peak shapes of basic compounds. At the same time, anionic chaotropicity contributes to the formation of ion pairs with cationic solutes, as well as the adsorption of ion pairs on stationary phases. It also plays an important role in the chromatographic retention of analytes. Therefore, the use of ILs as a mobile phase additive can greatly improve the analysis time, separation efficiency, resolution, and symmetry of basic compounds, which is attributed to the mixed mechanism involving hydrophobic partitioning, ion-pairing, and ion-exchange [50].

ILs can be used not only as mobile phase additives, but also for the modification/functionalization of stationary phases for GC [52] and LC [53]. Gas chromatography (GC) is widely used for the separation of volatile organic compounds and enantiomers because of its advantages of rapidity, sensitivity, simplicity, and accurate quantification. Chiral ionic liquids (CILs) have high thermal stability and can be used as chiral stationary phases (CSPs) in GC. Kimaru et al. synthesized a chiral ionic liquid, ([L-PheEtO][(C_2_F_5_SO_2_)_2_N]), which has high thermal stability and can be used as a stationary phase in GC. Its ability to recognize chirality was demonstrated using 2,2,2-trifluoroanthrylethanol (TFAE) as a model chiral analyte through fluorescence spectroscopy [54]. High-performance liquid chromatography (HPLC) has been proven to be one of the most widespread techniques for enantiomeric separation and analysis [55]. Rahim et al. reported new chiral stationary phases, (β-CD-BIMOTs, β-CD-DIMOTs), prepared using aromatic ionic liquids functionalized with β-cyclodextrin, designed to enhance the enantiomeric separation of flavonoids and β-blockers. They compared the enantiomeric separation ability of non-steroidal anti-inflammatory drugs (ibuprofen, indoprofen, ketoprofen, and fenoprofen) between β-CD-BIMOTs and β-CD-DIMOTs stationary phases with the native β-CD stationary phase, demonstrating that β-CD-BIMOTs stationary phases have better separation abilities due to the superposition of hydrogen bonding, hydrophobic, and also p–p interactions [56].

### 2.3. Drug Solubilization

The efficacy of pharmaceutical formulations is largely dependent on bioavailability, which is directly related to the solubility of the drug in aqueous or biological media and its permeability across biological membranes at body temperature [57]. Many drugs have good pharmacological activity, but they also have the disadvantage of poor solubility. About 40% of drugs are difficult to enter the stage of drug development due to poor solubility and permeability [2]. Traditionally, a number of strategies have been used to solve the problem of poor drug solubility, such as organic solubilizer, solid dispersions, lipid formulations, complexation with cyclodextrins, etc. [6]. ILs are liquid salts with a low melting point (usually below 100 °C), which, in some cases, can form an exotic room-temperature solvent that can solvate a variety of compounds well. This special solvation capacity enables ILs to solubilize many insoluble drugs, ultimately enhancing drug penetration through physiological barriers, thereby improving therapeutic efficacy [58,59]. Recently, ILs, as green and designable solvents, have been used to develop more efficient drug delivery systems to improve the solubility/loading of insoluble drugs. As early as 2008, the JAITELY research group reported that imidazole ILs have solubilizing effects and can act as drug reservoirs for both controlled and extended release [60]. The mixed solvents (water and ILs) solubilize the solute by disrupting the self-association of water and reducing the interfacial tension between the drug and the solvent medium to achieve the solubilizing effect of ILs [61]. In addition, ILs can affect the solubility and partition coefficient of drugs in the skin by forming intermolecular interactions with drugs, including hydrogen bonding, van der Waals forces, and Woods-on-Woods bonding, etc. The structural properties of ILs, including the electron distribution, alkyl side chains, the number of alkyl groups, and anionic species, affect their solubilizing effects. Based on the principle of “similar solubility”, the longer the alkyl chain in ILs, which represents greater hydrophobicity, the greater the solubility of hydrophobic drugs, and the lower the solubility of hydrophilic drugs [18]. Caparica et al. prepared eight different ILs by combining an imidazolium or cholinium cation with bromide or amino acid anions, including (2-hydroxyethyl)trimethylammonium phenylalaninate[Cho][Phe], (2-hydroxyethyl)trimethylammonium glycinate [Cho][Gly], 1-ethyl-3-methylimidazolium bromide[Emim][Br], 1-butyl 3-methylimidazolium bromide [Bmim][Br], 1-ethyl-3-methylimidazolium phenylalaninate [Emim][Phe], 1-ethyl-3-methylimidazolium glycinate [Emim][Gly], 1-butyl-3-methylimidazolium phenylalaninate [Bmim][Phe], and 1-butyl-3-methylimidazolium glycinate [Bmim][Gly]. The ability of these synthetic ILs formed by these different anions and cations to solubilize poorly soluble drugs, such as ferulic acid, caffeic acid, p-coumaric acids, and rutin, was then evaluated. The results showed that these synthetic ILs, except for the imidazole halogenated ILs (Emim][Br], [Bmim][Br]), and the [Cho][Br] salt, can effectively increase the solubility of the studied phenolic compounds even at low concentrations of these ILs (0.1%). Different combinations of cations/anions had an effect on drug solubility, with anions having a greater effect on drug solubility. In addition, amino acid ILs proved to be better solubility promoters, independent of choline- or imidazole-derived cations. Among the amino acids studied, glycine-based ILs improved higher solubility. Moreover, when the percentage of these ILs increased, the solubility of the active compounds also increased, suggesting that higher amounts of ILs can improve drug solubility to a greater extent. Finally, preparation stability experiments showed that all prepared preparations were stable, and that these synthetic ILs did not negatively affect the integrity of the emulsion [62]. Masumeh et al. developed an efficient and environmentally friendly co-dissolution method using choline-based ILs as novel green solvents to improve acetaminophen solubility in water. They studied the solubility of acetaminophen in aqueous choline lactate (ChLa) and choline bitartrate (ChBi) solutions under normal pressure and at temperatures of 298.15 K, 303.15 K, 308.15 K, and 313.15 K. It was found that the higher the co-solvent concentration and temperature, the greater the solubility of acetaminophen, and ChLa was observed to be more capable in increasing the solubility of acetaminophen. In addition, it was found that the interaction between ionic liquid and the drug was enhanced with an increase in cosolvent concentration (Figure 4) [63]. Viçosa et al. produced ultrafine rifampicin particles using an ionic liquid (1-ethyl 3-methyl imidazolium methyl-phosphonate) as an alternative solvent and a phosphate buffer as an antisolvent. Compared with the original drug, the solubility of the prepared rifampicin granules increased by 30%, and the dissolution rate increased by 83%, improving the problem of difficult dissolution of rifampicin [64].

### 2.4. Application of Ionic Liquids in Drug Crystal Engineering

The physicochemical properties of different crystalline forms of the same drug, such as solubility, dissolution rate, melting point, chemical stability and density, etc., vary greatly, which is an important factor affecting the production of pharmaceutical preparations and the in vivo bioavailability of active pharmaceutical ingredients. Therefore, drug polymorph modulation is considered to be one of the most important aspects of drug development [65,66,67]. Studies have shown that the use of ILs as crystallization media can effectively control crystal form and improve crystal habit, allowing for the acquisition of crystal forms or habits that cannot be obtained using traditional organic solvents, being of great help in improving the quality of active pharmaceutical ingredients and facilitating the post-processing of pharmaceutical preparations [68,69,70]. Martins et al. studied five different crystallization processes using pure or mixed imidazolium-based room temperature ionic liquids (RTILs) as a solvent and methanol as the cosolvent to obtain pure gabapentin form IV for the first time, a highly unstable polymorph (Figure 5). Satisfactory results were obtained for all five crystallization processes. Afterwards, the researchers used molecular dynamics simulations to explain the mechanism of obtaining pure gabapentin form IV. The use of RTILs favoring the formation of form IV was justified by GBP-RTIL interactions, as evident from the radial distribution functions (RDF). The simulation results showed a strong influence of H_(acidic (C4/C6mim))_-O_(carboxylate)_ interaction on driving the formation of form IV [71].

Drug crystal habits not only have an effect on solubility and dissolution rates but also affect properties such as flowability, tableting, and mechanical strength. Common drug crystallization habits include needle-like, rod-like, flake-like, and block-like forms, but the needle-like and flake-like crystallization habits have poor powder flow characteristics, which is not conducive to the production of drug preparations. Both crystallization techniques and solvents affect the habit or morphology of crystals and, thus, the dissolution properties of the drug. ILs have been applied as a novel crystallization medium in studies on the improvement and modification of drug crystal habits by improving morphology or reducing size. Studies have shown that reducing the crystal size can effectively increase the solubility and dissolution rate of insoluble substances [72,73]. Karthika et al. used imidazole-based ILs and PF_6_ anions to crystallize ibuprofen, producing needle-like crystals with high aspect ratios. They added several organic solvents to [bmim][PF_6_] to form hydrogen bonds with [bmim][PF_6_], which altered the interactions between [bmim][PF_6_] and the crystal plane of the solute, reduced the aspect ratio of ibuprofen crystals, and obtained the desirable crystal habit. The study found that when IL is bound to the solvent by hydrogen bonding, the interaction of the cation with the specific surface of the solute can be sufficiently altered to lead to a significant change in the aspect ratio of synthetic crystals [74]. Table 1 presents examples of applications of ionic liquids in the pharmaceutical field.

## 3. Ionic Liquids in Biomedical Applications

### 3.1. Drug Delivery Vehicle

Most drug delivery systems involve the process of getting drugs into the body for therapeutic effects through an appropriate technical route. Sometimes, a drug delivery system also plays a role in a certain diagnostic procedure. Conventional drug delivery systems (DDSs) always face significant challenges associated with unfavorable distribution and pharmacokinetics of drug molecules in vivo, resulting in low bioavailability and harmful systemic side effects. Researchers have been continuously trying to improve drug delivery systems to improve drug bioavailability, reduce drug loss, and prevent adverse reactions [75]. When drugs are not incorporated into the delivery vehicle, the concentration of the drug in the body rapidly reaches a supra-optimal level, followed by a rapid decline to a sub-optimal level due to systemic metabolism or degradation in the circulation, which result in the inadequate absorption of the drug at the target site [76,77]. For a long time, researchers have been investigating extended-release formulations that provide optimal therapeutic concentrations of drugs. Due to the chemical stability and biocompatibility of ILs, they have attracted much more attention from scholars, providing a feasible way to solve the problems faced by traditional DDSs [78,79,80,81]. In addition, the hydrophilicity and lipophilicity of ILs can be adjusted through the structural modification of anions and cations to improve the bioavailability of some drugs with low transdermal delivery efficiency due to solubility problems. At present, ILs with transdermal osmosis activity mainly include imidazoles, amino acid esters, choline, and so on. In recent years, naturally derived anions (such as amino acids, organic acids, and fatty acids) and cations (such as choline, amino acid ester, glycine betaine, and protein-derived cations) have been designed and prepared as ILs for application in DDSs [82,83,84]. Although this approach has certain effects and feasibility, it is still in the early stages of research.

Surfactant ionic liquids (SAILs) are a class of ionic liquids carrying short alkyl chains, which have a structure similar to that of cationic surfactants due to the presence of a charged hydrophilic head group and a non-polar hydrophobic tail group. They have attracted a lot of interest from researchers around the world [85,86,87]. The remarkable properties of SAIL, including high thermal stability and biodegradability, make it potentially useful as a drug delivery system [88,89,90]. Today, SAILs are used to design stable drug-carrying vehicles as surfactants or co-surfactants, including micelles, vesicles, microemulsion systems, microporous polymers, and active pharmaceutical ingredients (Figure 6) [77,91,92,93,94,95,96].

#### 3.1.1. Micelles

The use of SAIL-based micellar drug carriers is more advantageous than conventional surfactant-based drug carriers because of their smaller size, narrow size distribution, improved stability, and enhanced bioavailability of drugs for micelles [75]. SAILs are composed of hydrophilic charged head groups and hydrophobic long-chain alkyl groups, which can self-assemble to form nanostructures with adjustable physicochemical properties and biodegradability, and have been used to replace conventional surfactants in the preparation of micelles in a number of related studies [90,97,98]. Kumar et al. synthesized the ionic liquids 1-tetradecyl-3-methylimidazolium bromide [C_14_mim] [Br] and 1-pentadecyl-3-methylimidazolium bromide[C_15_mim] [Br] using 1-methylimidazole and alkyl halides 1-bromotetradecane and 1-bromopentadecane. Subsequently, they investigated the micellization behavior of surface-active ionic liquids [C_14_mim] [Br] and [C_15_mim] [Br] in the presence of the antidiabetic drug metformin hydrochloride, using conductivity measurements and infrared spectroscopy [99]. Ali et al. developed a SAIL-assisted nonaqueous micelle formulation using cholinium oleate (SAIL[Cho][Ole]) and sorbitan monolaurate (Span 20) as a transdermal drug delivery system for the delivery of paclitaxel (PTX). The SAIL-based micelle formulation significantly improved the solubility of PTX compared to the conventional tween-80-based micelle formulation (MF). The results showed that the formed micelles could improve the stability of the formulations while improving drug encapsulation efficiency and skin penetration. In vitro transdermal experiments using mouse skin as a model showed that the skin penetration performance of PTX was significantly improved in SAIL[Cho][Ole]/Span-20 (2:1)-based MF compared to the skin penetration performance of PTX in other MFs. The safety profile of the SAIL-based drug carriers was evaluated via in vivo histological analyses, demonstrating its potential as a safe nanoparticle for PTX transdermal delivery (Figure 7) [100].

#### 3.1.2. Microemulsions

Microemulsions (MEs), single optically isotropic and thermodynamically stable dispersions comprising oil, surfactant, and aqueous phases, have potential applications in the biomedical field due to their simplicity of preparation, long-term stability, biocompatibility, and high solubilization. They have become promising smart drug carriers [101,102,103]. MEs have the advantages of enhanced drug transdermal penetration ability, good solvation of the active drug, excellent thermodynamic stability, and easy preparation. Additionally, they can dissolve a variety of polar and nonpolar substances in their nanodomains. These advantages make them widely used in transdermal drug delivery [104]. Two types of microemulsions, water-in-oil (W/O) and oil-in-water (O/W), are widely used to develop effective delivery systems for polar and nonpolar drugs [105,106]. Mandal et al. reported the formulation of an ionic liquid containing a Tween-based nonaqueous microemulsion with biologically acceptable components, which indicates that the microemulsion droplets are spherical in nature as their size varies linearly with the increasing ionic liquid content. The nonaqueous IL/O microemulsion system can be used as a drug delivery system for many sparingly water-soluble drug molecules [105]. Deep eutectic compounds (DECs) usually consist of two or more components that can form hydrogen bonds with each other, resulting in a eutectic mixture [107]. Both ILs and DECs have special properties, including negligible vapor pressure, good solubility for many polar or non-polar compounds, and high thermal stability, making them widely used in the pharmaceutical field [108,109]. ILs are tunable design solvents that can be functional in all phases of the ME system by varying the composition of water, oil, and surfactant. MEs based on ILs have been tested as potential drug carriers for substance delivery [89]. Artemisinin is a sesquiterpene compound with a peroxide bridge used for the treatment of malaria. Zhang et al. proposed a microemulsion system based on an ionic liquid (1-hydroxyethyl-3-methylimidazole chloride) and deep eutectic compounds (lidocaine ibuprofen) to improve the transdermal delivery of artemisinin. In this microemulsion system, deep eutectic lidocaine ibuprofen with high artemisinin solubility was used as the oil phase, and the imidazole ionic liquid 1-hydroxyethyl-3-methylimidazole chloride ([HOEmim]Cl) with enhanced permeability was incorporated into the aqueous phase as a transdermal promoter. Lidocaine ibuprofen possessed excellent transdermal ability, biocompatibility, the ability to reduce malaria fever, and high solvation capability for Artemisinin. The optimized weight ratio of the microemulsion carrier was the water phase/the surfactant phase/the oil phase (45%:45%:10%) with an artemisinin loading of 1.0%. The surfactant phase contained polyoxyethylene sorbitan monooleate (Tween-80) and sorbitan monolaurate (Span-20), with ethanol (1:1:1) as a co-surfactant. Then, they performed in vitro transdermal experiments and found that artemisinin transport through the skin was significantly enhanced, with a 3-fold higher permeation flux than that of the isopropyl myristate system over 6 h. The results showed that the permeation flux of artemisinin through the skin was significantly enhanced using this microemulsion system. The effect of IL-based microemulsion (ILME) on stratum corneum (SC) was investigated using DSC, ATR-FTIR and AFM. The results show that ILME reduces the barrier of the SC by disrupting the regular arrangement of keratin, which enhances the transdermal delivery of artemisinin [110].

#### 3.1.3. Vesicles

Vesicles, unilamellar or multilamellar spheroid structures composed of amphiphilic molecules in an aqueous medium, are size-selective filters with a structure similar to that of biological membranes and can be used as models for biological membranes and in drug delivery. Vesicles are also called liposomes. Generally, if the amphiphilic molecules are synthetic surfactants, the structure formed is called a vesicle. However, if the amphiphilic molecules are natural surfactants such as lecithins, the structure formed is called a liposome. They can also be used as controlled-release carriers for active molecules such as drugs, flavors, nutrients, fragrances, and dyes [92,111]. Vesicles have the advantages of simple formulation, good chemical stability, and resistance to hydrolysis and oxidative degradation in a water medium [112]. When vesicles are used in the transdermal drug solubilization system, the toxicity of the drug is reduced, and the skin permeability of the drug is enhanced [113]. Jain et al. synthesized unilamellar anionic vesicles using SAILs (C_12_EMeImBr and C_12_EMorphBr) and the antiangiogenic agent sodium butyrate (NaBut). By encapsulating curcumin in these vesicles, its water solubility and stability can be improved to protect curcumin from the attack of inactivated enzymes in the biological solution and improve the medicinal and therapeutic effects of curcumin. Subsequently, the encapsulation efficiency of synthesized vesicles for the hydrophobic drug curcumin was studied. The result shows that the solubility of curcumin was increased from 0.0006 mg/mL to 0.80 mg/mL in morpholinium-based catanionic vesicles. The antibacterial activity of curcumin-encapsulated vesicles was further studied, and the maximum inhibitory zone against *Staphylococcus aureus* was 14.5, which confirmed the efficiency of curcumin-encapsulated vesicles [114].

#### 3.1.4. Microporous Polymers

In addition to micelles, microemulsions, and vesicles, there are also other types of drug carriers applied in drug delivery systems, such as microporous polymers. Encapsulating drugs into nanoparticles can protect the drug, control the rate of drug release, and deliver the targeted drug to specific tissues while increasing its bioavailability and reducing adverse side effects [115,116,117]. Microporous polymers can be used as effective drug delivery carriers due to their ability to adjust pore size, morphology, and surface properties. A variety of polymer nanoparticles have been used as drug carriers due to their good biocompatibility and degradability, showing the ability to enhance drug absorption, bioavailability, solubility, and stability [118,119,120,121]. Combining polymer nanoparticles with ILs can take advantage of the synergistic effect of the two materials to produce drug delivery systems with higher physicochemical and colloidal stability and increased drug loading, thereby improving the efficacy of poorly soluble drugs while reducing possible adverse side effects [122,123]. Júlio et al. produced an IL-polymer particle hybrid system composed of a choline-amino acid IL, [Cho][Phe] or [Cho][Glu], as solubility enhancers and either PLGA 50:50 or PLGA 75:25 as polymers to load rutin using an adapted W/O/W double emulsion technique. The synthesized rutin-loaded IL-polymer particle hybrid system had a particle size of 250–300 nm, a low polydispersity index, and a zeta potential of about −40 mV. The drug association efficiency of the developed system was up to about 76% in the presence of [Cho][Phe], and its sustained release of rutin was up to 72 h. Cytotoxicity studies showed that this delivery system was non-toxic to HaCat cells [124].

#### 3.1.5. Active Pharmaceutical Ingredient Ionic Liquids

Active pharmaceutical ingredients (APIs) possessing limited aqueous solubility are called class II drugs in the biopharmaceutical classification system. The new formulations of these drug molecules are slow to dissolve in biological fluids, resulting in inadequate and inconsistent systemic exposure, and thus, poor clinical efficacy [27]. A good strategy for solving this problem is to convert the drug molecules themselves into active pharmaceutical ingredient ionic liquids (API-ILs) [19,125]. API-ILs are designed by pairing drug-active cations and anions to design IL-based drugs with better solubility, which can effectively cross various barriers to reach target cells [126,127,128]. According to the different formation mechanisms, API-ILs can be divided into three types; (i) the first type is the direct binding of the API as an anion to the counterion through ionic bonding, which is common; (ii) the second type is the neutral API that forms ionic prodrugs through covalent bonds and then combines with counterions; (iii)the third type combines the first two ways to form dual API-ILs (Figure 8) [129,130]. Among these API-ILs types, novel cleavable linkers may be used for the second type, which is one of the future research directions.

Many studies have reported the transdermal applications of APIs-ILs due to their characteristics, which combine those of both API and ILs [131,132]. Moshikur et al. synthesized a series of MTX-ILs using methotrexate (MTX), a potential anticancer prodrug, and biocompatible ILs (choline and amino acid esters) as cations. The solubility of MTX-ILs was at least 5000 times higher than that of free MTX and two orders of magnitude higher than that of the sodium salt of MTX in water and simulated body fluids (phosphate-buffered saline, simulated gastric, and simulated intestinal fluids). In addition, a proline ethyl ester MTX prodrug had similar solubility to that of the MTX sodium salt, but had better antitumor activity in vitro [133]. Berton et al. explored the bioavailability of lidocaine through its incorporation into the ionic liquid lidocainium docusate ([Lid][Doc]) and the deep eutectic Lidocaine‧Ibuprofen (Lid‧Ibu). They dissolved each form of lidocaine in a vehicle cream and applied it topically to Spra-gue-Dawley rats to study the transdermal absorption of the two forms of lidocaine, which was then compared with crystalline salt lidocaine chloride ([Lid]Cl). The concentrations of lidocaine in plasma varied with the API-based creams applied. The experimental results showed that the systemic absorption of hydrogen-bonded deep eutectic Lid‧Ibu is the fastest and highest among three kinds of API-based creams [134]. Rupniewska et al. synthesized nine virtually non-toxic amino acid alkyl esters of ibuprofen [AAOR][Ibu]. Then they studied the binding behavior of ibuprofen-based ionic liquids with bovine serum albumin (BSA) through thermodynamic and molecular modeling studies. It was found that the ibuprofen-based ionic liquids did not affect the binding of ibuprofen to BSA, and the ibuprofen-based ionic liquids with large cations (such as L-phenylalanine ethyl ester ibuprofenate and L-phenylalanine propyl ester ibuprofenate) were able to bind efficiently at three different BSA binding sites, while those with smaller cations had the same binding sites as ibuprofen [135].

### 3.2. Antimicrobial Effects of Ionic Liquids

Bacteria are known to be inherently resistant to antibiotics, having molecular mechanisms to protect themselves. Antimicrobial resistance increases morbidity and mortality and has become a major concern in healthcare organizations. With more and more antibiotics being used around the world, bacteria have a greater chance of developing more complex resistance to these antibiotics. Bacterial resistance strategies render antibiotics ineffective against microorganisms, which has led to increased antibiotic dosages or changes in antibiotic classes in healthcare settings. Researchers have been continually searching for new and efficient antimicrobial agents to combat the emerging antibiotic resistance [136,137,138].

High designable and tunable ILs have many excellent unique physicochemical properties that make them promising in the fight against pathogenic bacteria [139]. Electrostatic interactions occur between the cationic structure of the ILs and the negatively charged bacterial cell wall, and the insertion of hydrophobic alkyl chains into the phospholipid bilayer causes the rupture of the hydrophobic structure of the bacterial cell membrane, resulting in the death of the bacteria cell. Studies have shown that the longer the alkyl side chain of ILs, the greater their toxic and antibacterial activity. In general, ILs with 10 to 14 carbon atoms in their side chains show the highest antimicrobial activity [140,141,142]. When the number of carbon atoms in the alkyl chain reaches 14, the ILs have the maximum toxicity. After this point, their toxicity no longer increases as the number of carbon atoms continues to increase. This was called the “cut-off” effect [143,144]. The electrostatic attraction between ILs and the bacterial cell membrane is the primary link for the antibacterial effect of ILs. Differences in the charge density of ILs can significantly affect the electrostatic interactions, and consequently, the antibacterial properties of ILs. Zheng et al. synthesized a series of mono- or bis-imidazolium ILs with different substituents, and found that bis-imidazolium ILs have stronger antibacterial properties, which is due to the large charge density of imidazole cation in bis-imidazolium ILs. The amount of charge affects the antibacterial activity of imidazolium salts; generally, the greater the charge, the stronger the antibacterial activity [145]. The mechanisms of antimicrobial action of conventional antibiotics can be categorized in several ways: interfering with bacterial cell wall synthesis, inhibiting protein or nucleic acid synthesis, interrupting the bacterial membrane structure, and inhibiting metabolic pathways/bacterial enzymes [146]. ILs can affect bacteria in a variety of ways, including interacting with bacterial membranes and walls, destabilizing proteins and enzymes, perturbing cell metabolism, and damaging DNA or biomembranes (Figure 9). ILs bind with DNA in a variety of ways, depending on their structure. Cation binding with the minor groove of DNA is the most common. Binding occurs primarily through electrostatic interactions between cationic head groups and DNA phosphate groups and can be supported by hydrophobic interactions between cationic side chains and nonpolar DNA bases (Figure 9C). Extracellular polymeric substances (EPSs) are composed of proteins, lipids, polysaccharides, and free DNA. They are a key component of biofilms produced by bacteria. EPSs build and maintain a stable biofilm structure that promotes adhesion to surfaces and ensures increased resistance to antimicrobial agents. Ionic liquids can directly affect the EPS structure by interfering with the extracellular matrix. The components that form EPSs can generate hydrogen bonds between themselves. The hydrophobic ionic liquid can relax the EPS structure and promote ILs deep into the aggregate (Figure 9D) [147]. The mechanism of antibacterial action of ILs is thought to be adsorption and disruption of negatively charged cell membranes, which also lead to cytoplasmic leakage, or increased membrane permeability of ILs, leading to the disruption of intracellular processes and proteins [128]. ILs can form holes in the bacterial membrane, thereby disrupting the bacterial membrane, altering the cell membrane phospholipid arrangement and membrane potential, and ultimately destroying the overall fluidity and viscoelasticity of the bacterial cell membrane [148]. These changes affect the biochemical and biophysical processes of cells, including recognition, trafficking, signaling, migration, adhesion, division, and mechanotransduction, which may ultimately lead to different effects that culminate in cell death in the form of apoptosis and necrosis [149]. The antimicrobial process of ILs consists of four stages. In the first step, the ionic liquid is adsorbed onto the cell membrane; subsequently, electrostatic interactions occur, leading to the inactivation of membrane proteins and the interaction of the ionic liquid with membrane phospholipids; then, the phospholipid bilayer is disrupted and disintegrated, and leakage of intracellular cytoplasm occurs; finally, cell wall destruction leads to cell lysis [8].

A variety of ILs have been shown to have antimicrobial properties, such as imidazolium-based ILs [150,151,152,153], ammonium-based ILs [154,155], phosphonium-based ILs [156,157], pyridinium-based ILs [158,159], pyrrolidinium-based ILs [160,161], piperidinium-based ILs [162,163], quinolinium-based ILs [164,165], cholinium-based ILs [166,167,168,169], etc. Kubis et al. synthesized a new family of RTILs: imidazolium salts with alkoxymethyl substituents in the cation, specifically 3-ethyl-1-alkoxymethylimidazolium bis(trifluoromethylsulfonyl)imides, using an environmentally friendly pathway. The antimicrobial properties of synthesized RTILs are closely related not only to the length of the alkyl chain but also to the species of the microbial strain. The results of antimicrobial activity tests showed that 1-dodecyloxymethyl-3-ethylimidazoliumbis(trifluoromethylsulfonyl)imide with the longest alkyl chain in the alkoxymethyl group exhibited high and selective activity primarily against Gram-positive *Staphylococcus* spp. [170]. Kumer et al. designed a novel ionic liquid, called ortho-toludinium carboxylate ionic liquids (OTILs), and synthesized it through the Brønsted acid-base neutralization reaction between ortho toluidine and carboxylic acids. After that, antibacterial screening of the synthesized OTILs was conducted using Gram-positive bacteria, including Bacillus cereus, *Staphylococcus aureus*, Sarcina lutea, and Bacillus subtilis, and gram-negative bacteria, including Escherichia coli, Salmonella typhi, Pseudomonas aeroginosa, and Shigella dysenteria, using the well diffusion method. Almost in the same way, antifungal testing of three pathogenic fungi (Aspergillus niger, Saccharomyces cerevisiae, and Candida albicans) was performed using the end-diffusion well method. The results revealed that the synthetic OTILs showed promising antibacterial activity against both bacteria and fungi [171].

### 3.3. Applications in Stabilizing Proteins

ILs are characterized by negligible vapor pressure, a low melting point, strong polarity, high stability, low toxicity, and high miscibility with other liquids. These properties of ILs can be easily adjusted by changing the composition of cations and anions, resulting in solvents with different chaotropic and kosmotropic properties, which are important for determining the water solubility and protein stability of solutes [7,172,173]. Nowadays, ILs have very important applications in biotechnology systems, such as solvent and co-solvent for enzymes, stabilizers for protein, etc. [174,175]. As highly complex biomolecules that widely exist in life processes, proteins must retain their secondary structural elements to remain active [176]. In biological systems, various covalent (such as hydrogen bonding) and non-covalent bond (such as electrostatic and hydrophobic interactions) interactions of proteins work together to maintain their normal function, which is very important for the structural functional integrity of bioactive conformations of proteins [177]. Proteins are in a conformational equilibrium, including folded, partially folded, and unfolded states in solution. However, the conformational equilibrium state can be broken when the thermodynamic state of the system, such as temperature, pressure, pH, or solvent composition changes [178,179,180]. When proteins are exposed to these factors, they gradually change or unfold in structure, forming protein aggregates over time, which can lead to the loss of function and activity of proteins and enzymes, severely limiting their applications in biopharmaceuticals [181].

Additives are frequently added to prevent the unfolding and aggregation of proteins without disrupting their secondary and tertiary structures [182,183]. Since proteins are stabilized by a balance between intramolecular interactions and interactions with the solvent environment, the use of biocompatible co-solvents can be considered to maintain protein stability [176]. In order to develop stable biological drugs, the search for biocompatible ILs as potential stabilizers for proteins has attracted great attention [184,185]. Imidazolium-based ILs and choline-based ILs as potential stabilizers for proteins have received the most attention in recent years, with choline-based ILs having become one of the most suitable candidates for stabilizing biopharmaceuticals due to their excellent biocompatibility and protein stability [175]. Khachatrian et al. investigated the effect of seven cholinium- and imidazolium-based ILs on the structure and stability of hen egg-white lysozyme to find quantitative correlations between the properties of the ILs and their impact on protein stability. The result shows that the effect of ILs on protein stability is related to their intermolecular interactions in aqueous solutions, and the stabilizing effect of ILs on lysozyme is usually correlated with their proton acceptor capacity. The interaction of ionic liquids with HEWL at standard temperatures was assessed by monitoring changes in protein fluorescence. Furthermore, the thermal stability of HEWL in aqueous solutions containing ionic liquids was determined using the circular dichroism spectroscopy (CD) method. In most cases, the most studied ILs have little effect on the fluorescence of lysozyme, according to fluorescence data, and no effect on its native secondary and tertiary structure at room temperature, according to CD data [186]. Kumari et al. studied the inhibiting effect of ammonium and choline-based ILs, including tetrabutylammonium methanesulfonate (TBAMS), tetrabutylammonium chloride (TBACl), and choline chloride (CholCl), on amyloid aggregation of superoxide dismutase 1 (SOD1). They found that both TBAMS and TBACl aqueous solutions, mainly acting on the nucleation step of fibrillation, can effectively inhibited SOD1 fibrillation, and the inhibitory effect was enhanced with increasing concentration. However, CholCl, acting at both the nucleation and elongation steps of fibrillation, inhibits the fibrillar aggregation of SOD1 at lower concentrations but has the opposite effect at higher concentrations. Thioflavin T (ThT) intensity measurements suggested that the order of the inhibitory effect of these ILs was TBACl > TBAMS > CholCl. Molecular docking and molecular dynamics simulation analyses determined the putative binding sites and differential binding of ionic liquids with different forms of SOD1 (Figure 10). Different types of interactions involved in the binding of ILs with SOD1 was observed using Ligplot(AutoDock 4.2 software) (Figure 10(A2–C2) and Discovery Studio v4.5 software (Figure 10(A3–C3)) [187].

## 4. Summary and Outlook

In recent years, research on and application of ILs in the field of pharmaceutical and biomedical industry have been gradually increasing due to their excellent chemical stability, thermal stability, structural designability, and biocompatibility. The world market of ionic liquids grew at an annual rate of 9.2% from 2016 to 2021 [188]. This review focuses on the applications of ionic liquids in the pharmaceutical and biomedical fields. Ionic liquids can be used as (i) green solvents or catalysts in drug synthesis; (ii) extraction solvents, mobile phase additives and reagents for the modification/functionalization of stationary phases in drug analysis; (iii) solubilizer for insoluble drugs; (iv) crystallization media for effective control of crystal morphology and improvement of crystal habit; (v) drug carriers such as micellar, microemulsions, vesicles, and microporous polymers; (vi) active pharmaceutical ingredient ionic liquids; (vii) antimicrobial agents; (viii) protein stabilizers. From the research results in recent years, ILs have great potential in the pharmaceutical and biomedical fields. With further research, ILs can be used in combination with other emerging materials to develop materials with better structural properties, thus solving more difficult problems in the life science and pharmaceutical fields.

Although ionic liquids have shown wide application prospects in many fields, their potential safety risks to ecosystems and human health cannot be ignored. The inherent properties of ionic liquids, such as high viscosity, possible toxicity, high price, and unknown environmental impact, limit their use in the actual production of drugs. Due to the short duration of appearance, wide variety, and imperfect understanding of the physicochemical properties of ILs, there are still many difficulties in completely replacing traditional organic solvents and meeting the requirements of existing drug regulatory regulations. At present, many studies on ionic liquids are still in their infancy, and there are no appropriate standards for the study of toxicity and safety of ionic liquids. Although research results show that ionic liquids have very low cytotoxicity, the toxicity evaluation at the cellular level is far from enough, and it is necessary to make a more comprehensive evaluation of their biotoxicity in animal experiments to ensure the reliability and safety of ionic liquids in the field of pharmacy. In order to overcome the above-mentioned problems, it is particularly important to select appropriate anions and cations to improve the toxicity of ionic liquids. It is best to select non-toxic anions and cations with good biocompatibility to prepare safer and effective ionic liquids, along with conducting a comprehensive biotoxicity evaluation.

ILs present many problems in terms of their synthesis, toxicity, and biodegradability, which can pose as great a risk to the environment as traditional solvents if used in large quantities and indiscriminately. Synthesis, recovery, toxicity, and biodegradability should be considered when using ionic liquids. The synthesis of ionic liquids involves the use of solvents that are highly hazardous to both human health and the environment, as well as the use of many organic compounds, such as volatile compounds containing N, S, halogens, and VOCs, which contribute to acid rain and smog formation [189]. The synthesis process should be improved for ionic liquids to be truly considered green from a synthesis point of view. Furthermore, when ionic liquids are used as catalysts in chemical synthesis processes, they should have low degradation rates and high chemical and thermal stability in order to reduce side reactions. At present, the release of alkylimidazole ionic liquids in the environment has been detected, and this pollution will be aggravated with the expansion of the application scale of ionic liquids in various fields in the future. Degradation of some long-chain alkyl alkylimidazole ionic liquids is difficult due to their highly stable chemical properties. It is a feasible strategy to synthesize new ILs using choline, pyridine, or morpholine cations instead of imidazole cations. At present, ester chains and anion chains, such as alkyl sulfate, alkyl sulfonate, and organic acid salts, perform well in terms of low toxicity and good biodegradability. The recovery and recycling process is an important aspect that must be considered when using ionic liquids. Efficient recovery and recycling measures can effectively reduce the production of ionic liquids, as well as their emissions in the environment. Although there are many problems to be solved, ILs still have great application potential in pharmaceuticals and biomedicine. Since the chemical structure of ionic liquids can be designed, it is an important research direction to develop safe, efficient, non-toxic, and environmentally friendly ionic liquids for medical applications.

## Figures and Tables

**Figure 1 pharmaceutics-16-00151-f001:**
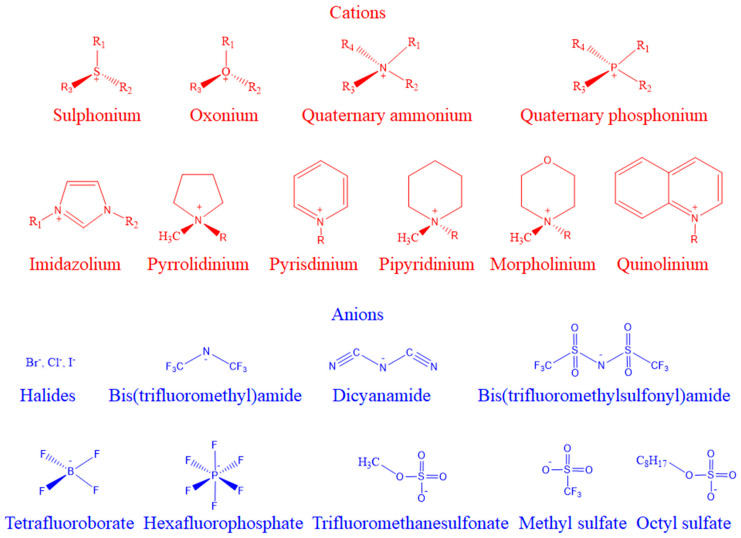
Cations and anions commonly used in ionic liquids. Reprinted with permission from Ref. [8]. Copyright (2021) John Wiley & Sons.

**Figure 2 pharmaceutics-16-00151-f002:**
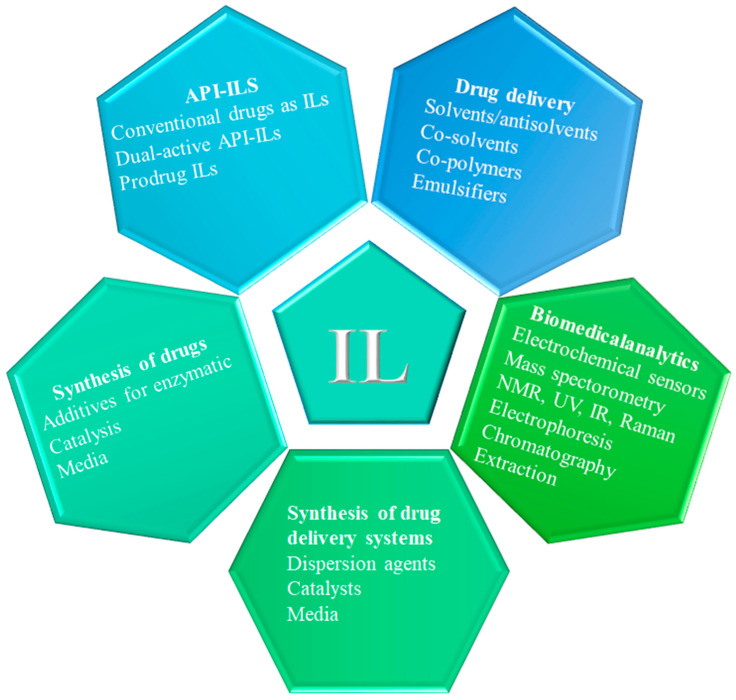
Possible applications of ionic liquids in pharmaceutics and biomedicine. Reprinted with permission from Ref. [18]. Copyright (2017) American Chemical Society.

**Figure 3 pharmaceutics-16-00151-f003:**
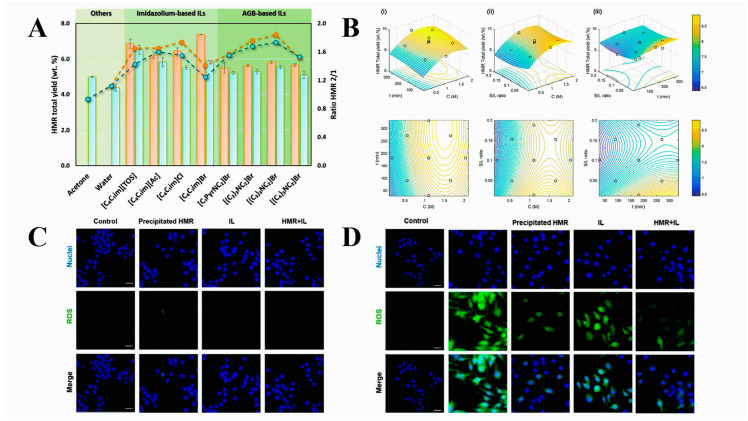
(**A**) Yield of HMR extracted with different aqueous solutions of ILs (at 0.5 M), acetone, and water (T = 25 °C, t = 180 min) for a solid–liquid (S/L) ratio = 0.10 (blue square) and for a S/L ratio = 0.02 (orange square). Ratio of HMR_2_/HMR_1_ for a S/L ratio = 0.10 (blue circle) or for a S/L ratio = 0.02 (orange circle). (**B**) Response surface and contour plots of the yield of total HMR extracted using aqueous solutions of [(C_2_)_3_NC_2_]Br at 25 °C with the combined effects of (i) extraction time (t) and IL concentration (C); (ii) solid–liquid ratio (S/L ratio) and concentration (C); and (iii) solid–liquid ratio (S/L ratio) and extraction time (t). (**C**) Macrophage cellular oxidative stress in the presence of different samples of HMR and pure IL. Scale bar: 20 µm. (**D**) Cellular oxidative stress in LPS-stimulated macrophages treated with the precipitated HMR, the HMR in IL solution, and pure IL. Scale bar: 20 µm. Reprinted with permission from Ref. [46]. Copyright (2017) Royal Society of Chemistry.

**Figure 4 pharmaceutics-16-00151-f004:**
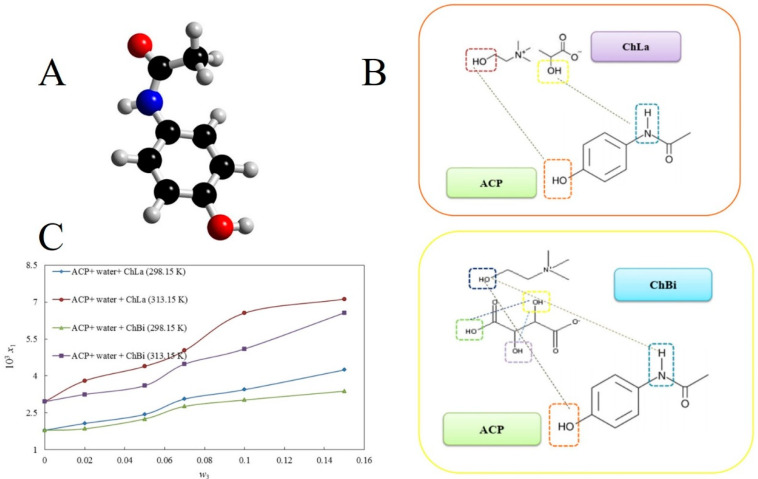
(**A**) Chemical structure of acetaminophen. (**B**) H-bonding interactions between the ACP and ILs. (**C**) The relationship between solubility of ACP (mol fraction x1) versus weight fraction of IL(w_3_) in aqueous solutions at 298.15 and 313.15 K; the solid lines are obtained from the e-NRTL model. Reprinted with permission from Ref. [63]. Copyright (2020) Elsevier B.V.

**Figure 5 pharmaceutics-16-00151-f005:**
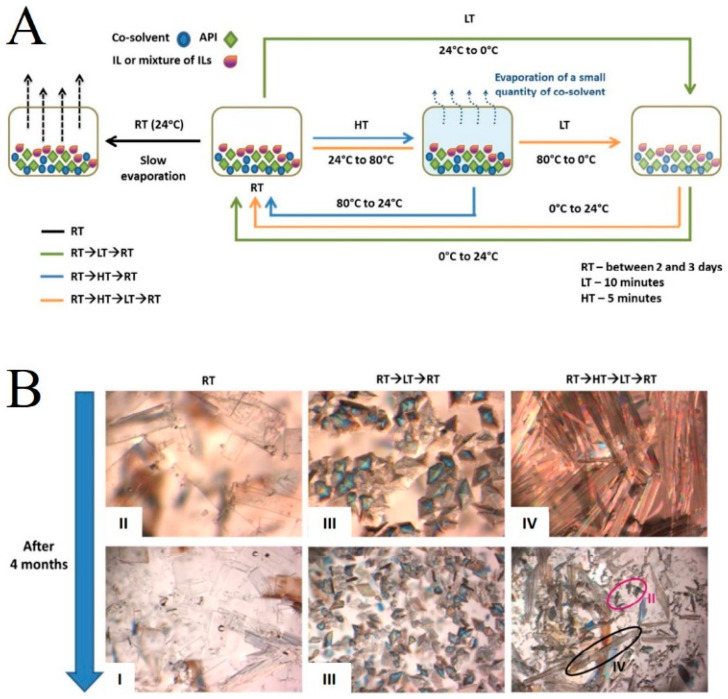
(**A**) Schematic diagram of the process of obtaining GBP crystals. RT, LT, and HT represent 24 °C, 0 °C, and 80 °C, respectively. (**B**) Crystal images of GBP polymorphic forms obtained after crystallization and 4 months later in C_4_mimBF_4_ + C_6_mimBF_4_ maintaining forms II or III over time, while form IV slowly converts to form II, forming a mixture (II + IV). Reprinted with permission from Ref. [71]. Copyright (2017) American Chemical Society.

**Figure 6 pharmaceutics-16-00151-f006:**
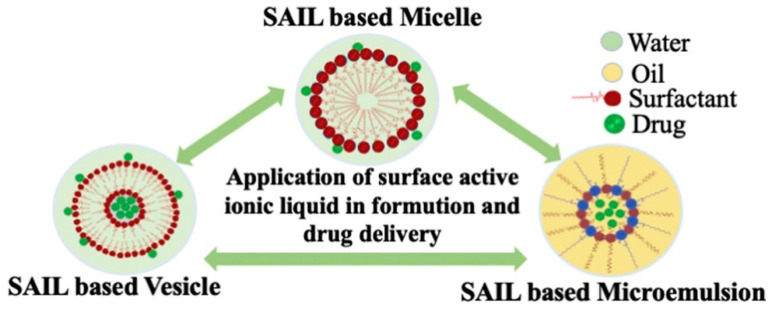
Schematic representation of three typical drug carriers. Reprinted with permission from Ref. [77]. Copyright (2021) Elsevier B.V.

**Figure 7 pharmaceutics-16-00151-f007:**
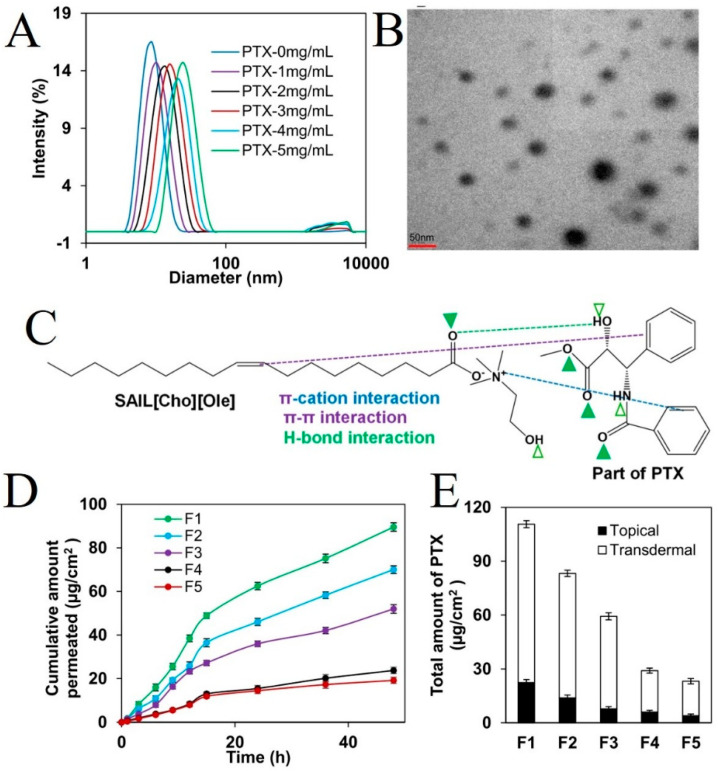
(**A**) The sizes and size distributions of micelles droplets in the SAIL[Cho][Ole]−based MF at 25 °C (DLS). (**B**) TEM images of the SAIL[Cho][Ole]−based MF (scale bar: 50 nm, 20 k resolution). (**C**) Schematic diagram showing the interactions between PTX and SAIL[Cho][Ole]/Span−20. The closed triangles represent the hydrogen bond acceptors, and the open triangles represent the hydrogen bond donors. (**D**) Transdermal permeation profiles of PTX delivered by various MFs with varying SAIL/co−surfactant ratios (S/Co). (**E**) Total (topical + transdermal) delivery of PTX by various MFs with varying S/Co ratios after 48 h (mean ± SD, *n* = 3). All the formulations contained 5 mg/mL PTX. F1 (50 mg surfactant mixture, SAIL[Cho][Ole]/Span−20 = 2:1); F2 (50 mg surfactant mixture, SAIL[Cho][Ole]/Span−20 = 1:1); F3 (50 mg surfactant mixture, SAIL[Cho][Ole]/Span−20 = 1:2); F4 (50 mg surfactant mixture, Tween 80/Span−20 =1:2); and F5 (100 mg ethanol). Reprinted with permission from Ref. [100]. Copyright (2021) American Chemical Society.

**Figure 8 pharmaceutics-16-00151-f008:**
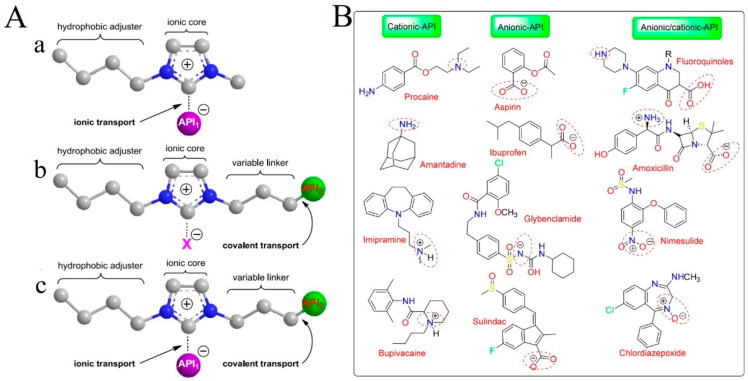
(**A**) Three types of API-ILs: API-IL containing ionic API as its anion (**a**), covalently linked API within its cation (**b**), and both binding options (**c**). Reprinted with permission from Ref. [130]. Copyright (2015) American Chemical Society. (**B**) Some promising API-ILs with good solubilization and penetration properties. Reprinted with permission from Ref. [19]. Copyright (2023) Multidisciplinary Digital Publishing Institute.

**Figure 9 pharmaceutics-16-00151-f009:**
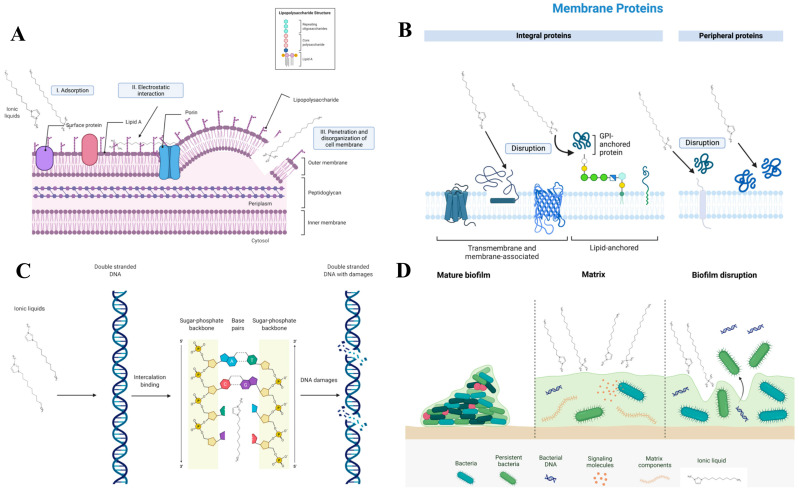
Antimicrobial mechanisms of ionic liquids. (**A**) Scheme of Gram-negative cell wall with membrane treated with ionic liquids. (**B**) Interactions of ionic liquids with proteins. (**C**) Interactions of ionic liquids with DNA structure. (**D**) Biofilm disruption after interaction with ionic liquids. Reprinted with permission from Ref. [147]. Copyright (2023) Elsevier B.V.

**Figure 10 pharmaceutics-16-00151-f010:**
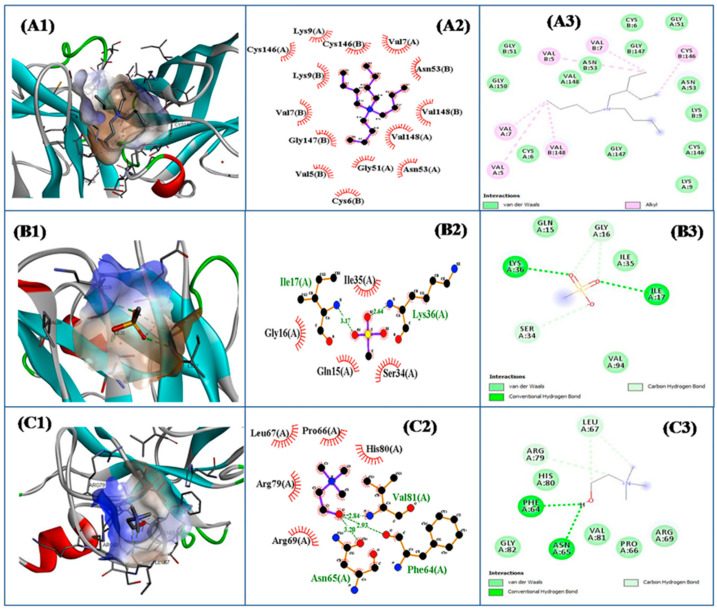
Lowest energy-binding model of SOD1-IL complex: SOD1- [tetrabutylammonium cation] complex (**A1**), SOD1-[methanesulfonate anion] complex (**B1**) and SOD1-[cholinium cation] complex (**C1**). Different type of interactions involved in binding of ILs with SOD1 ((**A2**,**B2**,**C2**) and (**A3**,**B3**,**C3**). Hydrogen bonds and hydrophobic interactions are shown in green and red, respectively. Reprinted with permission from Ref. [187]. Copyright (2022) Elsevier B.V.

**Table 1 pharmaceutics-16-00151-t001:** Examples of pharmaceutical applications of ILs.

No.	Compound	Activity	ILs	Role of IL	Ref.
1	Curcumin diacetate	Antioxidant, anticarcinogenic, etc.	Bis(trifluoromethylsulfonyl)imide-based ILs	Reaction medium	[30]
2	1,8-dioxooctahy-droxanthene derivative	Anticancer	1-butyl-3-methylimidazoliumtetrafluoroborate, 1-butyl-3-methylimidazoliumbromide, 1-butyl-3-methylimidazoliumchloride	Reaction medium	[31]
3	Imidazole derivatives	Antioxidant	Pyrrolidinium hydrogen sulfate	Catalyst	[33]
4	4-methoxyphenol	Drug intermediate	1,3-disulfonic acid imidazolium hydrogen sulfate	Catalyst	[35]
5	Diarylmethanes	Drug intermediate	1-propylsulfonic acid-3-methylimidazolium trifluoromethanesulfonate	Catalyst	[36]
6	7-hydroxymatairesinol	Anticarcinogenic and antioxidative	Glycine-betaine ILs	Extractant	[46]
7	Carbamazepine	Anticonvulsant	(Z)-octadec-9-en-1-aminium tetrachloroferrate (III), (Z)-octadec-9-en-1-aminium trichlorocobaltate (II)	Adsorbents	[47]
8	Jatrorrhizine, palmatine, berberine	Antiinflammation, antihypertension, etc.	1-hexyl-3-methylimidazolium tertafluoroborate	additives	[50]
9	Ibuprofen, indoprofen, ketoprofen, fenoprofen	Anti-inflammatory	β-cyclodextrin functionalized ILs	stationary phase	[56]
10	Ferulic acid, caffeic acid, p-coumaric acids, rutin	Antioxidant, antimicrobial, etc.	[Cho][Phe], [Cho][Gly], [Emim][Br], [Bmim][Br], [Emim][Phe], [Emim][Gly], [Bmim][Phe], [Bmim][Gly]	Solubilizing agent	[62]
11	Acetaminophen	Analgesic, antipyretic	Choline-based ILs	Solubilizing agent	[63]
12	Ultrafine rifampicin particles	Antibacterial, antiviral	1-ethyl 3-methyl imidazolium methyl-phosphonate	Solvent	[64]
13	Gabapentin form IV	Neuroleptic	Imidazolium-based ILs	Crystallization directing agents	[71]
14	Crystallized ibuprofen	Analgesic, antipyretic	Imidazole-based ILs	Crystallization directing agents	[74]

## Data Availability

Details are available from authors.

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
