# Peer review of "Ionic Liquids in Pharmaceutical and Biomedical Applications: A Review"

_pharmaceutics, 2024, doi:10.3390/pharmaceutics16010151_

Round 1

Reviewer 1 Report

Comments and Suggestions for Authors

The article gives a nice overview about different fields of applications ionic liquids by describing specific examples, despite some weakness is that not always a general picture of the meaning of these examples is presented. The reviewer has the following comments:

Lines 44-46: Sentence should be revised: “stable in….”. The second part of the sentence is also not conclusive and possibly incomplete: “which is …”

Lines 53-54: Statement makes no sense: “which also makes it alive research in bio medicine is increasing gradually”

Page 2: the two paragraphs are somehow repetitive, as both end up mentioning the chemical composition. It is recommended to start off with explaining the physicochemical properties of ionic liquids compared to other solvents, then introducing the general concept of chemical structures of IL and finally categorizing them in different generations.

Line 70: It seems relevant to distinguish between experimental exploration of IL in biomedicine and the real use in pharmaceutical products or medical treatments.

Page 4, application of IL as solvents for syntheses: The authors give some examples were IL have been used in synthesis. However, readers may be interested to understand the more general picture on which basis a certain IL can be selected for a specific synthesis. What are the selection criteria? Which types of reactions (reaction mechanisms) work in which IL?

Page 4, purification: As IL are not volatile, the purification procedure could also be considered in the discussion of synthesis strategies, as it may differ from conventional procedures.

Lines 129-130: “ILs also has certain catalytic and selective effects on some reactions”. It would be informative to readers, if groups of conventional catalysts/reactions may be mentioned, which could be substituted. Subsequently, very detailed examples are given, but what might be useful for readers exploring the field is also a more general picture. Please also correct the typo: has ->have.

Lines 134: Check grammar: act -> acting?

Lines 143: Check grammar: which is highly efficient and green catalytic systems, for the….

Page 5, section 2.2: The headline is “analytics”, but the text is about production technology to isolate secondary metabolites via extraction from natural sources. 

Line 191: recommend to introduce a new paragraph before: “Dispersive solid phase extraction ….” This is, where analytical topics start to be discussed.

Lines 198-201: It remains unclear: i) what is meant with “hydrophilic perimeter”, ii) what is the advantage to have a magnetic IL, and iii) which detection method is used, as this method is not mentioned (only extraction method is explained).

Fig. 4: It may be helpful to add to the legend of Panel B, that extraction time and IL concentration were varied (if this is, what is shown). Similarly, an assignment of staining in panel C and D should be provided.

Lines 228-229: It should be part of this review to briefly present the proposed mechanism of IL in chromatographic separation.

Lines 256-259: These statements may be misunderstood. Using ethanol as cosolvent is not a problem in many cases, while methanol certainly is. However, no qualified pharmacist would use methanol as a cosolvent in a drug product. It may be relevant to distinguish between solvents used in drug product manufacturing (which should subsequently be removed if toxic) and solubilizers, which are selected based on non-toxicity and are part of the formulation.

Line 227: remove underlining: “Caparica et al. prepared”

Line 280-285: It may be interesting to discuss structure function relationships, just listing the chemical names of the different solvents is not very informative.

Line 295: typo:  Water -> water

Line 312: pharmaceutical active ingredient -> active pharmaceutical ingredient

Lines 318-327:  Not clear why it is advantageous to obtain an unstable crystal modification. Use dashes instead of dots in Lines 324/326.

Lines 337: Fluidity may not be the right term. The authors may mean: powder flow characteristics

Lines 359-360: “There are four major types of drug delivery systems, namely oral, injection-based, transdermal, and vector-based.” This statement is incorrectly mixing up application routes (e.g. oral) and delivery systems (namely carriers: e.g. tablets, nanoparticles, plasters, vectors). The reviewer also does not agree about that there a four major drug delivery systems.

Lines 431-432: Statement seems incomplete: “MEs have the advantages of enhanced drug transdermal ……. ability”. A word may be missing, such as “transportation”, “penetration”, “permeation” etc.

Lines 441-442: Not clear what the term multifunctional means in this context: “ILs are tunable design solvents that can be multifunctional in all phases of the ME system”

Lines 446-457: Additional information may be added to explain the therapeutic concept of solubilizing artemisinine in a lidocaine-ibuprofen eutectic mixture.

Lines 459-460: Check grammar: “the closed bilayers assembled unilamellar or multilamellar spheroid structures”

Lines 467-468: Typo “catan-ionic“; „antiangio-genic”

Line 510: It may be relevant to note that cleavable inkers are needed for case (ii).

Fig. 9: Figure and legend should be plotted on the same page.

All section 3: The biological fate of IL used for parenteral application should be discussed.

Section 3.3: The authors only discuss alkyl chain length, but did not comment on charge effects.

Lines 580-583: Dots should be used at the end of the sentences.

Fig.10: Size is to small and plotting quality is insufficient to allow readability of structural formulas and labels. The effects on DNA and biofilms as shown in panel C and D are not justified by citations and explanations in the text. Typical antimicrobial substances that affect cell membrane integrity do not necessarily intercalate with DNA.

Lines 644-646: In previous sentences, the stabilizing effect is discussed, while here the word “destabilization” is used. Is this a typo? Also, the correlation of fluorescence characteristics and tertiary structure may not be obvious to all readers based on the phrasing that is used.

Lines 673-681: It is not clear, why the pharmaceutical field and the biomedical field are separated in this listing. It is also not clear, while topics such as drug delivery systems APIs or antimicrobial agents are assigned to the biomedical, rather than the pharmaceutical field.

Lines 685-700: It may be inappropriate to handle all types of IL similarly when speaking about toxicity. Some differentiation may be helpful as toxicity is related to the specific chemical compositions rather than the general categorization of a substance as ionic liquid.

Comments on the Quality of English Language

Expressions are not always clear, see specific comments

Author Response

Dear editor,

We are truly grateful to yours and reviewers’ critical comments and thoughtful suggestions. Based on these comments and suggestions, we have made careful modifications on the original manuscript. All changes made to the text are in red color. We hope the new manuscript will meet your magazine’s standard. Below you will find our point-by-point responses to the reviewers’ comments/questions:

Comments and Suggestions for Authors

The article gives a nice overview about different fields of applications ionic liquids by describing specific examples, despite some weakness is that not always a general picture of the meaning of these examples is presented. The reviewer has the following comments:

Lines 44-46: Sentence should be revised: “stable in….”. The second part of the sentence is also not conclusive and possibly incomplete: “which is …”

√ Many thanks for your suggestion. We greatly appreciate your kind thoughts. We have corrected the mistake.

Lines 53-54: Statement makes no sense: “which also makes it alive research in bio medicine is increasing gradually”

√ Many thanks for your suggestion. We greatly appreciate your kind thoughts. We have perfected the sentence.

Page 2: the two paragraphs are somehow repetitive, as both end up mentioning the chemical composition. It is recommended to start off with explaining the physicochemical properties of ionic liquids compared to other solvents, then introducing the general concept of chemical structures of IL and finally categorizing them in different generations.

√ Many thanks for your suggestion. We greatly appreciate your kind thoughts. We have adjusted the content of the introduction to make it more logical. Thank you very much!

Line 70: It seems relevant to distinguish between experimental exploration of IL in biomedicine and the real use in pharmaceutical products or medical treatments.

√ Many thanks for your suggestion. We greatly appreciate your kind thoughts. In the subsequent sections of this paper, we have addressed the pharmaceutical and biomedical applications of ionic liquids, respectively.

Page 4, application of IL as solvents for syntheses: The authors give some examples were IL have been used in synthesis. However, readers may be interested to understand the more general picture on which basis a certain IL can be selected for a specific synthesis. What are the selection criteria? Which types of reactions (reaction mechanisms) work in which IL?

√ Many thanks for your suggestion. We greatly appreciate your kind thoughts. We have not gathered information on a more general method of selecting ionic liquids as synthetic solvents.

Page 4, purification: As IL are not volatile, the purification procedure could also be considered in the discussion of synthesis strategies, as it may differ from conventional procedures.

√ Many thanks for your suggestion. We greatly appreciate your kind thoughts. In the reference literature, the catalytic effect of ionic liquid was emphasized, and the purification procedure was not described.

Lines 129-130: “ILs also has certain catalytic and selective effects on some reactions”. It would be informative to readers, if groups of conventional catalysts/reactions may be mentioned, which could be substituted. Subsequently, very detailed examples are given, but what might be useful for readers exploring the field is also a more general picture. Please also correct the typo: has ->have.

√ Many thanks for your suggestion. We greatly appreciate your kind thoughts. We have not collected more information about this part. We have corrected the mistake.

Lines 134: Check grammar: act -> acting?

√ Many thanks for your suggestion. We greatly appreciate your kind thoughts. We have corrected the mistake.

Lines 143: Check grammar: which is highly efficient and green catalytic systems, for the….

√ Many thanks for your suggestion. We greatly appreciate your kind thoughts. We have corrected the mistake.

Page 5, section 2.2: The headline is “analytics”, but the text is about production technology to isolate secondary metabolites via extraction from natural sources. 

√ Many thanks for your suggestion. We greatly appreciate your kind thoughts. We have perfected the headline of section 2.2.

Line 191: recommend to introduce a new paragraph before: “Dispersive solid phase extraction ….” This is, where analytical topics start to be discussed.

√ Many thanks for your suggestion. We greatly appreciate your kind thoughts. We have taken your suggestion.

Lines 198-201: It remains unclear: i) what is meant with “hydrophilic perimeter”, ii) what is the advantage to have a magnetic IL, and iii) which detection method is used, as this method is not mentioned (only extraction method is explained).

√ Many thanks for your suggestion. We greatly appreciate your kind thoughts. During the extraction process, the sorbent forms micelles in water with a hydrophobic core and hydrophilic perimeter. The hydrophilic perimeter refers to the hydrophilic end of the micelles. We have added this premise in the article. And the advantage of magnetic ILs and the detection method are also supplemented in the article.

Fig. 4: It may be helpful to add to the legend of Panel B, that extraction time and IL concentration were varied (if this is, what is shown). Similarly, an assignment of staining in panel C and D should be provided.

√ Many thanks for your suggestion. We greatly appreciate your kind thoughts. We have added the legend of panel B and the assignment of staining in panel C and D.

Lines 228-229: It should be part of this review to briefly present the proposed mechanism of IL in chromatographic separation.

√ Many thanks for your suggestion. We greatly appreciate your kind thoughts. We have added the proposed mechanism of IL in chromatographic separation in the article.

Lines 256-259: These statements may be misunderstood. Using ethanol as cosolvent is not a problem in many cases, while methanol certainly is. However, no qualified pharmacist would use methanol as a cosolvent in a drug product. It may be relevant to distinguish between solvents used in drug product manufacturing (which should subsequently be removed if toxic) and solubilizers, which are selected based on non-toxicity and are part of the formulation.

√ Many thanks for your suggestion. We greatly appreciate your kind thoughts. We have removed the misleading expression. Thank you very much!

Line 227: remove underlining: “Caparica et al. prepared”

√ Many thanks for your suggestion. We greatly appreciate your kind thoughts. We have removed the underlining.

Line 280-285: It may be interesting to discuss structure function relationships, just listing the chemical names of the different solvents is not very informative.

√ Many thanks for your suggestion. We greatly appreciate your kind thoughts. We have added the information about structure function relationships. Thank you very much!

Line 295: typo:  Water -> water

√ Many thanks for your suggestion. We greatly appreciate your kind thoughts. We have corrected the mistake.

Line 312: pharmaceutical active ingredient -> active pharmaceutical ingredient

√ Many thanks for your suggestion. We greatly appreciate your kind thoughts. We have corrected the mistake.

Lines 318-327:  Not clear why it is advantageous to obtain an unstable crystal modification. Use dashes instead of dots in Lines 324/326.

√ Many thanks for your suggestion. We greatly appreciate your kind thoughts. It is not easy to obtain pure gabapentin form IV due to its highly unstable. In this reference, the pure GBP Form IV was isolated through RTILs for the first time. We have used dashes instead of dots.

Lines 337: Fluidity may not be the right term. The authors may mean: powder flow characteristics

√ Many thanks for your suggestion. We greatly appreciate your kind thoughts. We have corrected the mistake.

Lines 359-360: “There are four major types of drug delivery systems, namely oral, injection-based, transdermal, and vector-based.” This statement is incorrectly mixing up application routes (e.g. oral) and delivery systems (namely carriers: e.g. tablets, nanoparticles, plasters, vectors). The reviewer also does not agree about that there a four major drug delivery systems.

√ Many thanks for your suggestion. We greatly appreciate your kind thoughts. We are very sorry for the mistake, and the wrong information has been deleted. Thank you very much!

Lines 431-432: Statement seems incomplete: “MEs have the advantages of enhanced drug transdermal ……. ability”. A word may be missing, such as “transportation”, “penetration”, “permeation” etc.

√ Many thanks for your suggestion. We greatly appreciate your kind thoughts. We have corrected the mistake.

Lines 441-442: Not clear what the term multifunctional means in this context: “ILs are tunable design solvents that can be multifunctional in all phases of the ME system”

√ Many thanks for your suggestion. We greatly appreciate your kind thoughts. We have corrected the inaccurate word.

Lines 446-457: Additional information may be added to explain the therapeutic concept of solubilizing artemisinine in a lidocaine-ibuprofen eutectic mixture.

√ Many thanks for your suggestion. We greatly appreciate your kind thoughts. We have added the information to explain the therapeutic concept of solubilizing artemisinine in a lidocaine-ibuprofen eutectic mixture.

Lines 459-460: Check grammar: “the closed bilayers assembled unilamellar or multilamellar spheroid structures”

√ Many thanks for your suggestion. We greatly appreciate your kind thoughts. We have corrected the mistake.

Lines 467-468: Typo “catan-ionic“; antiangio-genic”

√ Many thanks for your suggestion. We greatly appreciate your kind thoughts. We have corrected the mistake.

Line 510: It may be relevant to note that cleavable linkers are needed for case (ii).

√ Many thanks for your suggestion. We greatly appreciate your kind thoughts. And the thought has been added to the manuscript as below, “Among these API-ILs types, novel cleavable linkers may be used for the second type, which is one of the future research directions.” Thank you very much!

Fig. 9: Figure and legend should be plotted on the same page.

√ Many thanks for your suggestion. We greatly appreciate your kind thoughts. We have taken note of this issue. Thank you very much! 

All section 3: The biological fate of IL used for parenteral application should be discussed.

√ Many thanks for your suggestion. We greatly appreciate your kind thoughts. We have collected no information on the biological fate of IL used for parenteral application.

Section 3.3: The authors only discuss alkyl chain length, but did not comment on charge effects.

√ Many thanks for your suggestion. We greatly appreciate your kind thoughts. We have collected no information on the charge effect at this time.

Lines 580-583: Dots should be used at the end of the sentences.

√ Many thanks for your suggestion. We greatly appreciate your kind thoughts. We have corrected the mistake.

Fig.10: Size is to small and plotting quality is insufficient to allow readability of structural formulas and labels. The effects on DNA and biofilms as shown in panel C and D are not justified by citations and explanations in the text. Typical antimicrobial substances that affect cell membrane integrity do not necessarily intercalate with DNA.

√ Many thanks for your suggestion. We greatly appreciate your kind thoughts. This view is quoted from reference 139, which reviews the possible antibacterial mechanism of ionic liquids as novel antimicrobial agents.

Lines 644-646: In previous sentences, the stabilizing effect is discussed, while here the word “destabilization” is used. Is this a typo? Also, the correlation of fluorescence characteristics and tertiary structure may not be obvious to all readers based on the phrasing that is used.

√ Many thanks for your suggestion. We greatly appreciate your kind thoughts. We have corrected the mistake. In addition, we have added supplementary information on fluorescence characteristics and tertiary structure. Thank you very much!

Lines 673-681: It is not clear, why the pharmaceutical field and the biomedical field are separated in this listing. It is also not clear, while topics such as drug delivery systems APIs or antimicrobial agents are assigned to the biomedical, rather than the pharmaceutical field.

√ Many thanks for your suggestion. We greatly appreciate your kind thoughts. We have corrected the statement.

Lines 685-700: It may be inappropriate to handle all types of IL similarly when speaking about toxicity. Some differentiation may be helpful as toxicity is related to the specific chemical compositions rather than the general categorization of a substance as ionic liquid.

√ Many thanks for your suggestion. We greatly appreciate your kind thoughts. This paragraph is trying to convey is the current lack of sufficient assessment of the potential toxicity and safety of ionic liquids. It the end, we have pointed out that it is best to select non-toxic anions and cations with good biocompatibility to prepare safer and effective ionic liquids and conduct a comprehensive biotoxicity evaluation.

Yours sincerely

Prof. Yong-Gang Zhao

College of Biological and Environmental Engineering, Zhejiang Shuren University, Hangzhou, 310015, China

E-mail address: [email protected]

Reviewer 2 Report

Comments and Suggestions for Authors

The work is really interesting and significant. However, the structure of the paper is completely wrong, as well as there are many terminological errors. Indeed when I got to the beginning of chapter 3.2. I realized that the structure absolutely cannot be like that, because what is at the beginning of 3.2. you write the whole chapter 3.1. I really think a major revision is needed. To be more clear, serious work restructuring and logical connection.

1. Consider switching the order of the first and second paragraphs in the introduction. Somehow, it is more logical for me to go for a general division of ionic liquids, a division by generations... and then a specific application in pharmacy, biomedicine, and in general which generation would that be? To be more precise, I would transfer that first paragraph about the application in pharmacy later after the general description, where you mention it again.

2. Line 175, consider whether the term therapeutic agents is correct, perhaps rather a compound with biological activity...

3. Part 2.3. not only chemical modifications are made to increase solubility/dissolution rate/permeability and therefore bioavailability. This is also done by the wording, please correct it. For example. solid dispersions, lipid formulations, complexation with cyclodextrins...

4. 267 and 268 line reference 53, please check the terminology modified, controlled, extended release.

5. In chapter 2 the whole, especially 2.3. and 2.4. try to give more examples and specific significant results that have been achieved. As you mentioned reference 67 (lines 351-355), just think ie. check that this reference and this segment have a place in chapter 2.3. or 2.4. you state that the dissolution rate and solubility are increased (which is chapter 2.3). Nowhere in this example have you listed the characteristics of the polymorphic form?

6. Row 358, I don't like the term therapeutic effect... It is true that drug delivery systems have a medicinal substance in 99% of cases, which will have a therapeutic effect, but sometimes they also play a role in a certain diagnostic procedure.

7. Line 359 is by no means an expression of the four types of systems, this is the four ways of applying the medicine..... And this vector is even a little questionable... But these are not types of systems, but ways of application.

8. Reference 68, consider the term drug deprivation

9. Read lines 365-375 it seems that no formulation approach other than ionic liquids has produced the desired results. It's not like that. Ionic liquids have their own positive effects, give examples, but do not rule out other approaches, they have their own formulations on the drug market that are specifically used and give excellent results in therapy.

10. Please state if there is any formulation with ionic liquids on the market or in the clinical research phase and what are the results, what is the clinical research stage (you will understand that my question under 9 makes sense). Don't be too superlative about your topic, but a little more realistic comparison.

11. Lines 377-385 have a lot of repetition...

12. Line 413 chapter title micelles and you talk about nanoparticles. It is absolutely not the same. Maybe this comment of mine is harsh, but somehow the whole paper seems to me technologically written, with a weaker application in pharmacy and medicine. It seems to me, the first time I looked, that none of the authors are pharmacists, possibly doctors. That's why you simply have such terminological errors. You did not give too many applicable examples. Errors in the type of route of administration and type of therapeutic systems. Some sentences are taken from papers and those in the context where one sentence is taken have a completely different meaning. There is no strong connection, especially when we talk about wording.

13. References 98.99 Read a little, clarify, and expand. This sounds weird

14. Are microemulsions used only for skin application? There is no single system for oral administration and i.v. application? First... There are also registered preparations, especially self-micro emulsifying systems.

15. Row 443 in microemulsions, reference 83, nanosystems for substance delivery. Please read everything again and we don't have such illogicalities.

16. What is the drop size of microemulsion systems?

17. Please add the pharmacopoeial and chemical names in brackets for Tween and Span, the journal is from the pharmaceutical group.

18. 3.1.3. vesicular systems have a wide range of systems, try to be more specific

19. Line 479, 480 this first sentence is unacceptable nanosystems and microemulsions.... see the division... Microemulsions have a smaller drop size than nanoemulsions. Everything is mixed, please be careful.

20. Order 497 extended-release. Who? What? Medicinal substances. Who is? We can't just extend the release. The terminology is not aligned in your work and in a previous question, I mentioned a problem with the term extended-release.

21. Where did the beginning of 3.2 come from? Well, you wrote about the problem of poor solubility and permeability in 3.1. about solubilization and now again. Not really.

Comments on the Quality of English Language

 Minor editing of English language required

Author Response

Dear editor,

We are truly grateful to yours and reviewers’ critical comments and thoughtful suggestions. Based on these comments and suggestions, we have made careful modifications on the original manuscript. All changes made to the text are in red color. We hope the new manuscript will meet your magazine’s standard. Below you will find our point-by-point responses to the reviewers’ comments/questions:

Comments and Suggestions for Authors

The work is really interesting and significant. However, the structure of the paper is completely wrong, as well as there are many terminological errors. Indeed when I got to the beginning of chapter 3.2. I realized that the structure absolutely cannot be like that, because what is at the beginning of 3.2. you write the whole chapter 3.1. I really think a major revision is needed. To be more clear, serious work restructuring and logical connection.

√ Many thanks for your suggestion, and we greatly appreciate your kind suggestion.

  1. Consider switching the order of the first and second paragraphs in the introduction. Somehow, it is more logical for me to go for a general division of ionic liquids, a division by generations... and then a specific application in pharmacy, biomedicine, and in general which generation would that be? To be more precise, I would transfer that first paragraph about the application in pharmacy later after the general description, where you mention it again.

√ Many thanks for your suggestion. We greatly appreciate your kind thoughts. We have adjusted the content of the introduction to make it more logical. Thank you very much!

  1. Line 175, consider whether the term therapeutic agents is correct, perhaps rather a compound with biological activity...

√ Many thanks for your suggestion. We greatly appreciate your kind thoughts. We have corrected the mistake.

  1. Part 2.3. not only chemical modifications are made to increase solubility/dissolution rate/permeability and therefore bioavailability. This is also done by the wording, please correct it. For example. solid dispersions, lipid formulations, complexation with cyclodextrins...

√ Many thanks for your suggestion. We greatly appreciate your kind thoughts. We have corrected the mistake.

  1. 267 and 268 line reference 53, please check the terminology modified, controlled, extended release.

√ Many thanks for your suggestion. We greatly appreciate your kind thoughts. We have corrected the mistake.

  1. In chapter 2 the whole, especially 2.3. and 2.4. try to give more examples and specific significant results that have been achieved. As you mentioned reference 67 (lines 351-355), just think ie. check that this reference and this segment have a place in chapter 2.3. or 2.4. you state that the dissolution rate and solubility are increased (which is chapter 2.3). Nowhere in this example have you listed the characteristics of the polymorphic form?

√ Many thanks for your suggestion. We greatly appreciate your kind thoughts. We have moved reference 67 to chapter 2.3.

  1. Row 358, I don't like the term therapeutic effect... It is true that drug delivery systems have a medicinal substance in 99% of cases, which will have a therapeutic effect, but sometimes they also play a role in a certain diagnostic procedure.

√ Many thanks for your suggestion. We greatly appreciate your kind thoughts. We have corrected the mistake. Thank you very much!

  1. Line 359 is by no means an expression of the four types of systems, this is the four ways of applying the medicine..... And this vector is even a little questionable... But these are not types of systems, but ways of application.

√ Many thanks for your suggestion. We greatly appreciate your kind thoughts. We are very sorry for the mistake, and the wrong information has been deleted. Thank you very much!

  1. Reference 68, consider the term drug deprivation

√ Many thanks for your suggestion. We greatly appreciate your kind thoughts. We have deleted the term deprivation.

  1. Read lines 365-375 it seems that no formulation approach other than ionic liquids has produced the desired results. It's not like that. Ionic liquids have their own positive effects, give examples, but do not rule out other approaches, they have their own formulations on the drug market that are specifically used and give excellent results in therapy.

√ Many thanks for your suggestion. We greatly appreciate your kind thoughts. We have corrected this misleading expression.

  1. Please state if there is any formulation with ionic liquids on the market or in the clinical research phase and what are the results, what is the clinical research stage (you will understand that my question under 9 makes sense). Don't be too superlative about your topic, but a little more realistic comparison.

√ Many thanks for your suggestion. We greatly appreciate your kind thoughts. We have not collected more information about this part. And we will keep it in our mind in the further study. Thank you very much!

  1. Lines 377-385 have a lot of repetition...

√ Many thanks for your suggestion. We greatly appreciate your kind thoughts. We have simplified this paragraph.

  1. Line 413 chapter title micelles and you talk about nanoparticles. It is absolutely not the same. Maybe this comment of mine is harsh, but somehow the whole paper seems to me technologically written, with a weaker application in pharmacy and medicine. It seems to me, the first time I looked, that none of the authors are pharmacists, possibly doctors. That's why you simply have such terminological errors. You did not give too many applicable examples. Errors in the type of route of administration and type of therapeutic systems. Some sentences are taken from papers and those in the context where one sentence is taken have a completely different meaning. There is no strong connection, especially when we talk about wording.

√ Many thanks for your suggestion. We greatly appreciate your kind thoughts. We have corrected the mistake.

  1. References 98.99 Read a little, clarify, and expand. This sounds weird

√ Many thanks for your suggestion. We greatly appreciate your kind thoughts. We have corrected the wrong expression.

  1. Are microemulsions used only for skin application? There is no single system for oral administration and i.v. application? First... There are also registered preparations, especially self-micro emulsifying systems.

√ Many thanks for your suggestion. We greatly appreciate your kind thoughts. We have removed inappropriate range restrictions about the application of microemulsion.

  1. Row 443 in microemulsions, reference 83, nanosystems for substance delivery. Please read everything again and we don't have such illogicalities.

√ Many thanks for your suggestion. We greatly appreciate your kind thoughts. We have corrected the mistake.

  1. What is the drop size of microemulsion systems?

√ Many thanks for your suggestion. We greatly appreciate your kind thoughts. The radius of the microemulsion droplet is usually 10-100 nm.

  1. Please add the pharmacopoeial and chemical names in brackets for Tween and Span, the journal is from the pharmaceutical group.

√ Many thanks for your suggestion. We greatly appreciate your kind thoughts. We have added the information about Tween and Span.

  1. 3.1.3. vesicular systems have a wide range of systems, try to be more specific

√ Many thanks for your suggestion. We greatly appreciate your kind thoughts. We have added more information about vesicular systems.

  1. Line 479, 480 this first sentence is unacceptable nanosystems and microemulsions.... see the division... Microemulsions have a smaller drop size than nanoemulsions. Everything is mixed, please be careful.

√ Many thanks for your suggestion. We greatly appreciate your kind thoughts. We have corrected the mistake.

  1. Order 497 extended-release. Who? What? Medicinal substances. Who is? We can't just extend the release. The terminology is not aligned in your work and in a previous question, I mentioned a problem with the term extended-release.

√ Many thanks for your suggestion. We greatly appreciate your kind thoughts. We have corrected the mistake.

  1. Where did the beginning of 3.2 come from? Well, you wrote about the problem of poor solubility and permeability in 3.1. about solubilization and now again. Not really.

√ Many thanks for your suggestion. We greatly appreciate your kind thoughts. To some extent, both 3.1 and 3.2 address the problem of poor solubility of drugs. 3.1 focuses on the role of the IL-based vesicles, micelles, and microemulsions as drug carrier in improving the solubility of poorly soluble drugs. However, API-ILs in 3.2 are designed by pairing drug-active cations and anions to design IL-based drugs with better solubility. Considering the different mechanism of action of ionic liquids in 3.1 and 3.2, the two cases are described separately. Thank you very much!

Yours sincerely

Prof. Yong-Gang Zhao

College of Biological and Environmental Engineering, Zhejiang Shuren University, Hangzhou, 310015, China

E-mail address: [email protected]

Reviewer 3 Report

Comments and Suggestions for Authors

The proposed review seems to me well organized and proposes in each section at least one example illustrating the application or the reason for the section. The use of ionic liquids that improve the performance of bioactive compounds is an important handicap in this type of compounds.

Again, in this type of review, I miss a methodology that helps to avoid repeating the contents. From my point of view, there is no guide/index on how to proceed in the review, and as I have mentioned, neither is the methodology used. All the figures proposed in the review are reprinted with editorial permissions, but there is no figure of the authors themselves, nor a table summarizing the contents, so there is a lack of originality. Some figures, such as figure 2, could be improved and made their own. Figure 3 is full of feasibility studies that do not look good. Finally, I do not know why antibiosis appears as a key word if it is not mentioned.

Only two typos were detected:

- "zhang et al." Line 444

- hyphen in the word "antigio-genic" line 468

This is considered an adequate revision that needs to respond to the above suggestions for publication.

Author Response

Dear editor,

We are truly grateful to yours and reviewers’ critical comments and thoughtful suggestions. Based on these comments and suggestions, we have made careful modifications on the original manuscript. All changes made to the text are in red color. We hope the new manuscript will meet your magazine’s standard. Below you will find our point-by-point responses to the reviewers’ comments/questions:

Comments and Suggestions for Authors

The proposed review seems to me well organized and proposes in each section at least one example illustrating the application or the reason for the section. The use of ionic liquids that improve the performance of bioactive compounds is an important handicap in this type of compounds.

√ Many thanks for your suggestion, and we greatly appreciate your kind suggestion.

Again, in this type of review, I miss a methodology that helps to avoid repeating the contents. From my point of view, there is no guide/index on how to proceed in the review, and as I have mentioned, neither is the methodology used.

√ Many thanks for your suggestion. We greatly appreciate your kind thoughts. The outline of this review has been added to the manuscript as below. Thank you very much!

Contents

  1. Introduction
  2. Ionic Liquids in Pharmaceutical Applications

2.1.    Application of Ionic Liquid in Drug Synthesis

2.2.    Application of Ionic Liquid in Drug Extraction and Analysis

2.3.    Drug Solubilization

2.4.    Application of Ionic Liquids in Drug Crystal Engineering

  1. Ionic Liquids in Biomedical Applications

3.1.    Drug Delivery Vehicle

3.1.1.    Micelles

3.1.2.    Microemulsions

3.1.3.    Vesicles

3.1.4.    Microporous Polymers

3.2.    Active Pharmaceutical Ingredient Ionic Liquids

3.3.    Antimicrobial Effects of Ionic Liquids

3.4.    Applications in Stabilizing Proteins

  1. Summary and Outlook

Acknowledgments

References

All the figures proposed in the review are reprinted with editorial permissions, but there is no figure of the authors themselves, nor a table summarizing the contents, so there is a lack of originality. Some figures, such as figure 2, could be improved and made their own.

√ Many thanks for your suggestion. We greatly appreciate your kind thoughts. Figure 1 and figure 2 have been redrawn as below. Table 1 presents examples of applications of ionic liquids in the pharmaceutical field. Thank you very much!

Figure 1. Cations and anions commonly used in ionic liquids. Reprinted with permission from Ref. [8]. Copyright (2021) John Wiley & Sons.

Figure 2. Possible applications of ionic liquids in pharmaceutics and biomedicine. Reprinted with permission from Ref. [18]. Copyright (2017) American Chemical Society.

Table 1 examples of pharmaceutical applications of ILs

No.

Compound

Activity

ILs

Role of IL

Ref

1

Curcumin diacetate

Antioxidant, anticarcinogenic, etc.

Bis(trifluoromethylsulfonyl)imide-based ILs

Reaction medium

[30]

2

1,8-dioxooctahy-droxanthene derivative

Anticancer

1-butyl-3-methylimidazoliumtetrafluoroborate,

1-butyl-3-methylimidazoliumbromide,

1-butyl-3-methylimidazoliumchloride

Reaction medium

[31]

3

Imidazole derivatives

Antioxidant

Pyrrolidinium hydrogen sulfate

Catalyst

[33]

4

4-methoxyphenol

Drug intermediate

1,3-disulfonic acid imidazolium hydrogen sulfate

Catalyst

[35]

5

Diarylmethanes

Drug intermediate

1-propylsulfonic acid-3-methylimidazolium trifluoromethanesulfonate

Catalyst

[36]

6

7-hydroxymatairesinol

Anticarcinogenic and antioxidative

Glycine-betaine ILs

Extractant

[39]

7

Carbamazepine

Anticonvulsant

(Z)-octadec-9-en-1-aminium tetrachloroferrate (III), (Z)-octadec-9-en-1-aminium trichlorocobaltate (II)

Adsorbents

[40]

8

Jatrorrhizine, palmatine, berberine

Antiinflammation, antihypertension, etc.

1-hexyl-3-methylimidazolium tertafluoroborate

additives

[43]

9

Ibuprofen, indoprofen, ketoprofen, fenoprofen

Antiinflammatory

β-cyclodextrin functionalized ILs

stationary phase

[49]

10

Ferulic acid, caffeic acid, p-coumaric acids, rutin

Antioxidant, antimicrobial, etc.

[Cho][Phe], [Cho][Gly], [Emim][Br], [Bmim][Br], [Emim][Phe], [Emim][Gly], [Bmim][Phe], [Bmim][Gly]

Solubilizingagent

[55]

11

Acetaminophen

Analgesic, antipyretic

Choline-based ILs

Solubilizing agent

[56]

12

Ultrafine rifampicin particles

Antibacterial, antiviral

1-ethyl 3-methyl imidazolium methyl-phosphonate

Solvent

[57]

13

Gabapentin form IV

Neuroleptic

Imidazolium-based ILs

Crystallization directing agents

[64]

14

Crystallize ibuprofen

Analgesic, antipyretic

Imidazole-based ILs

Crystallization directing agents

[67]

Figure 3 is full of feasibility studies that do not look good.

√ Many thanks for your suggestion. We greatly appreciate your kind thoughts. Figure 3 has been deleted. Thank you very much!

Finally, I do not know why antibiosis appears as a key word if it is not mentioned.

√ Many thanks for your suggestion. We greatly appreciate your kind thoughts. We have changed antibiosis to antimicrobial effect.

Only two typos were detected:

 "zhang et al." Line 444

√ Many thanks for your suggestion. We greatly appreciate your kind thoughts. We have corrected the mistake.

- hyphen in the word "antigio-genic" line 468

√ Many thanks for your suggestion. We greatly appreciate your kind thoughts. We have corrected the mistake.

This is considered an adequate revision that needs to respond to the above suggestions for publication.

√ Many thanks for your suggestion.

Yours sincerely

Prof. Yong-Gang Zhao

College of Biological and Environmental Engineering, Zhejiang Shuren University, Hangzhou, 310015, China

E-mail address: [email protected]

Round 2

Reviewer 1 Report

Comments and Suggestions for Authors

The reviewer was provided with a blank version of the manuscript in which changes were not visible. Thus it was not possible to perform a proper analysis of the revised manuscript.

From the response letter it seems, that the authors primarily concentrated on minor text edits, while the more general questions were not addressed. In the reviewers opinion, an excellent review paper should not just rephrase the content of other papers, but should extract ("synthesize") new knowledge by bringing all the information of individual papers together. One example for this is the removal of IL that are used as processing aids, which certainly is an important topics. Another example is the suitability of IL as solvents in certain types of chemical reactions, i.e. the question which reaction mechanisms are compatibile with which IL. 

Comments on the Quality of English Language

The reviewer has spend many hours on a detailed reading and provision of comments on grammar during the first review. As there was no document versions with marked changes provided, the improvement of language could not be checked in a timely manner.

Author Response

Dear reviewer,

We are truly grateful to your critical comments and thoughtful suggestions. Based on these comments and suggestions, we have made careful modifications on the original manuscript. All changes made to the text are in red color. We hope the new manuscript will meet your magazine’s standard. Below you will find our point-by-point responses to your comments/questions:

Comments and Suggestions for Authors

The reviewer was provided with a blank version of the manuscript in which changes were not visible. Thus it was not possible to perform a proper analysis of the revised manuscript. From the response letter it seems, that the authors primarily concentrated on minor text edits, while the more general questions were not addressed. In the reviewers opinion, an excellent review paper should not just rephrase the content of other papers, but should extract ("synthesize") new knowledge by bringing all the information of individual papers together. One example for this is the removal of IL that are used as processing aids, which certainly is an important topics. Another example is the suitability of IL as solvents in certain types of chemical reactions, i.e. the question which reaction mechanisms are compatibile with which IL. 

√ Many thanks for your suggestion, and we greatly appreciate your kind suggestion. We are very sorry for our previous responses to your comments/questions. In this version, we have carefully reviewed the comments/questions and have revised the manuscript accordingly. Thank you very much!

Comments and Suggestions for Authors

The article gives a nice overview about different fields of applications ionic liquids by describing specific examples, despite some weakness is that not always a general picture of the meaning of these examples is presented. The reviewer has the following comments:

Lines 44-46: Sentence should be revised: “stable in….”. The second part of the sentence is also not conclusive and possibly incomplete: “which is …”

√ Many thanks for your suggestion. We greatly appreciate your kind thoughts. We have corrected the mistake.

Lines 53-54: Statement makes no sense: “which also makes it alive research in bio medicine is increasing gradually”

√ Many thanks for your suggestion. We greatly appreciate your kind thoughts. We have perfected the sentence.

Page 2: the two paragraphs are somehow repetitive, as both end up mentioning the chemical composition. It is recommended to start off with explaining the physicochemical properties of ionic liquids compared to other solvents, then introducing the general concept of chemical structures of IL and finally categorizing them in different generations.

√ Many thanks for your suggestion. We greatly appreciate your kind thoughts. We have adjusted the content of the introduction to make it more logical. Thank you very much!

Line 70: It seems relevant to distinguish between experimental exploration of IL in biomedicine and the real use in pharmaceutical products or medical treatments.

√ Many thanks for your suggestion. We greatly appreciate your kind thoughts. In the subsequent sections of this paper, we have addressed the pharmaceutical and biomedical applications of ionic liquids, respectively.

Page 4, application of IL as solvents for syntheses: The authors give some examples were IL have been used in synthesis. However, readers may be interested to understand the more general picture on which basis a certain IL can be selected for a specific synthesis. What are the selection criteria? Which types of reactions (reaction mechanisms) work in which IL?

√ Many thanks for your suggestion. We greatly appreciate your kind thoughts. We have added this section: 2.1.2 Reaction Mechanism Analysis of Ionic Liquids in Drug Synthesis as below, “Imidazolium cation can react with Pd complexes to form carbene-Pd, which in some cases serves as a good catalyst for Heck or Suzuki reactions [37]. Mo et al. found that Heck arylation of electron-rich olefins could be accomplished with a variety of aryl bromides and iodides in excellent regioselectivity using the ionic liquid 1-butyl-3-methylimidazolium tetrafluoroborate ([bmim][BF4]) as a solvent without the need for aryl triflates or halide scavengers. Since ILs are composed entirely of ions, electrostatic interaction is more likely to produce a pd-olefin cation and a halide anion from two neutral precursors than a neutral pd-olefin intermediate from the same pre-cursor. They concluded that the use of ILs as solvents could facilitate the ionic pathway by generating branched olefins without the need for halide scavengers. Moreover, the acidic C2-H proton of the imidazolium ring forms hydrogen bonds with halide anions, which may also contribute to accelerating the ionic reaction by promoting the dissocia-tion of halide anions from palladium and its stabilization [38]. As we all know, the organic esters are very important pharmaceutical intermediates. Protophilic amide ionic liquid (PAIL) had already been used for esterification, and it can stabilize the intermediates in the esterification process. Xu et al. investigated the ester-ification reaction assisted by PAIL and its catalytic mechanism. The key two steps in the esterification reaction are the protonation of acid and the nucleophilic attack of the protonated acid by the alcohol. PAIL accelerated the esterification reaction by stabilizing the protonated acetic acid by sharing its electron cloud with protonated carbonyl and limiting the alcohol protonation. The addition of inorganic acids, the conventional cat-alyst, increased the chemical shift of carbon on the carbonyl of acetic acid, whereas PAIL did not have this effect. In [DMF]+HSO4−, the acetic acid was not protonated by H+, but attacked by [DMF]+, forming new stable intermediates. The carbonyl of DMF shares electrons with the protonated carbonyl group of acetic acid, giving it a higher electron density than the protonated carbonyl group in inorganic acids. In addition, PAIL is less able to protonate alcohols than inorganic acids. Therefore, the alcohols in PAIL attack the protonated carbonyl group of acetic acid more readily than the alcohols in inorganic acids. Based on these two aspects, PAIL accelerates the esterification reaction [39]. ILs can promote the formation of active reaction substrates during reactions. Akbari et al. synthesized a-aminophosphonates from aldehydes and ketones using a sulfonic acid functionalized IL as a Brønsted acid catalyst. The reaction mechanism is the for-mation of active imine induced by IL. The subsequent reaction of the active amine with the added phosphite gives the phosphonium intermediate. Finally, the phosphonium intermediate reacts with water generated during imine formation to give a-aminophosphonate and methanol. ILs also contribute to some hydrogenation reactions [40]. Nian et al. studied the catalytic hydrogenation of citronellal to menthol using Cu/ZrO2-SiO2 as a catalyst in IL. The results showed that the cations in the ILs formed hydrogen bonds with the carbonyl group in the citronellal molecule, which made it easier to isomerize citronellal to huimenthol. In particular, the acidity-adjustable [bmim][AlmCln] ILs effectively provided the Lewis acid conditions for isomerization of citronellal, which promoted the conversion of citronellal toward the production of menthol in the competitive hydrogenation [41]. The Knoevenagel condensation is one of the most useful carbon–carbon bond forming reactions in organic syntheses. Xin et al. investigated the Knoevenagel con-densation reaction of various aromatic aldehydes with active methylene compounds using the synthesized cyclic guanidinium lactate ionic liquid as medium at room tem-perature in a high yield of over 90% in 1-7 min. It was found that the substitution group of the aromatic aldehyde, either the electron donating or withdrawing groups, had little effect on the reaction. And the activity of methylene was the key factor influencing the reaction speed [42]. Saha et al. synthesized 2-aryl benzimidazoles through o-phenylendiamine and several substituted aromatic aldehyde condensation reaction promoted by ionic liquid, [pmim]BF4. The nature and position of substituent on the aryl ring were found to have little effect on reactivity. The IL acts as both catalyst and reaction medium in the reaction. Firstly, [pmim]BF4 activates aldehyde and exposes it to the nucleophilic attack of o-phenylenediamine, which forms a monoaldimine. Subsequently, cyclization of the monoaldehyde imine was followed by oxidative dehydrogenation in air to generate benzimidazole [43].”

    Thank you very much!

Page 4, purification: As IL are not volatile, the purification procedure could also be considered in the discussion of synthesis strategies, as it may differ from conventional procedures.

√ Many thanks for your suggestion. We greatly appreciate your kind thoughts. We have supplemented the content on purification, which is shown in lines 112-122, lines142-145 and lines166-170.

Lines 129-130: “ILs also has certain catalytic and selective effects on some reactions”. It would be informative to readers, if groups of conventional catalysts/reactions may be mentioned, which could be substituted. Subsequently, very detailed examples are given, but what might be useful for readers exploring the field is also a more general picture. Please also correct the typo: has ->have.

√ Many thanks for your suggestion. We greatly appreciate your kind thoughts. We have added more information on ionic liquids as catalysts, displayed in section 2.1.2. We have corrected the mistake.

Lines 134: Check grammar: act -> acting?

√ Many thanks for your suggestion. We greatly appreciate your kind thoughts. We have corrected the mistake.

Lines 143: Check grammar: which is highly efficient and green catalytic systems, for the….

√ Many thanks for your suggestion. We greatly appreciate your kind thoughts. We have corrected the mistake.

Page 5, section 2.2: The headline is “analytics”, but the text is about production technology to isolate secondary metabolites via extraction from natural sources. 

√ Many thanks for your suggestion. We greatly appreciate your kind thoughts. We have perfected the headline of section 2.2.

Line 191: recommend to introduce a new paragraph before: “Dispersive solid phase extraction ….” This is, where analytical topics start to be discussed.

√ Many thanks for your suggestion. We greatly appreciate your kind thoughts. We have taken your suggestion.

Lines 198-201: It remains unclear: i) what is meant with “hydrophilic perimeter”, ii) what is the advantage to have a magnetic IL, and iii) which detection method is used, as this method is not mentioned (only extraction method is explained).

√ Many thanks for your suggestion. We greatly appreciate your kind thoughts. During the extraction process, the sorbent forms micelles in water with a hydrophobic core and hydrophilic perimeter. The hydrophilic perimeter refers to the hydrophilic end of the micelles. We have added this premise in the article. And the advantage of magnetic ILs and the detection method are also supplemented in the article.

Fig. 4: It may be helpful to add to the legend of Panel B, that extraction time and IL concentration were varied (if this is, what is shown). Similarly, an assignment of staining in panel C and D should be provided.

√ Many thanks for your suggestion. We greatly appreciate your kind thoughts. We have added the legend of panel B and the assignment of staining in panel C and D.

Lines 228-229: It should be part of this review to briefly present the proposed mechanism of IL in chromatographic separation.

√ Many thanks for your suggestion. We greatly appreciate your kind thoughts. We have added the proposed mechanism of IL in chromatographic separation in the article.

Lines 256-259: These statements may be misunderstood. Using ethanol as cosolvent is not a problem in many cases, while methanol certainly is. However, no qualified pharmacist would use methanol as a cosolvent in a drug product. It may be relevant to distinguish between solvents used in drug product manufacturing (which should subsequently be removed if toxic) and solubilizers, which are selected based on non-toxicity and are part of the formulation.

√ Many thanks for your suggestion. We greatly appreciate your kind thoughts. We have removed the misleading expression. Thank you very much!

Line 227: remove underlining: “Caparica et al. prepared”

√ Many thanks for your suggestion. We greatly appreciate your kind thoughts. We have removed the underlining.

Line 280-285: It may be interesting to discuss structure function relationships, just listing the chemical names of the different solvents is not very informative.

√ Many thanks for your suggestion. We greatly appreciate your kind thoughts. We have added the information about structure function relationships. Thank you very much!

Line 295: typo:  Water -> water

√ Many thanks for your suggestion. We greatly appreciate your kind thoughts. We have corrected the mistake.

Line 312: pharmaceutical active ingredient -> active pharmaceutical ingredient

√ Many thanks for your suggestion. We greatly appreciate your kind thoughts. We have corrected the mistake.

Lines 318-327:  Not clear why it is advantageous to obtain an unstable crystal modification. Use dashes instead of dots in Lines 324/326.

√ Many thanks for your suggestion. We greatly appreciate your kind thoughts. It is not easy to obtain pure gabapentin form IV due to its highly unstable. In this reference, the pure GBP Form IV was isolated through RTILs for the first time. We have used dashes instead of dots.

Lines 337: Fluidity may not be the right term. The authors may mean: powder flow characteristics

√ Many thanks for your suggestion. We greatly appreciate your kind thoughts. We have corrected the mistake.

Lines 359-360: “There are four major types of drug delivery systems, namely oral, injection-based, transdermal, and vector-based.” This statement is incorrectly mixing up application routes (e.g. oral) and delivery systems (namely carriers: e.g. tablets, nanoparticles, plasters, vectors). The reviewer also does not agree about that there a four major drug delivery systems.

√ Many thanks for your suggestion. We greatly appreciate your kind thoughts. We are very sorry for the mistake, and the wrong information has been deleted. Thank you very much!

Lines 431-432: Statement seems incomplete: “MEs have the advantages of enhanced drug transdermal ……. ability”. A word may be missing, such as “transportation”, “penetration”, “permeation” etc.

√ Many thanks for your suggestion. We greatly appreciate your kind thoughts. We have corrected the mistake.

Lines 441-442: Not clear what the term multifunctional means in this context: “ILs are tunable design solvents that can be multifunctional in all phases of the ME system”

√ Many thanks for your suggestion. We greatly appreciate your kind thoughts. We have corrected the inaccurate word.

Lines 446-457: Additional information may be added to explain the therapeutic concept of solubilizing artemisinine in a lidocaine-ibuprofen eutectic mixture.

√ Many thanks for your suggestion. We greatly appreciate your kind thoughts. We have added the information to explain the therapeutic concept of solubilizing artemisinine in a lidocaine-ibuprofen eutectic mixture.

Lines 459-460: Check grammar: “the closed bilayers assembled unilamellar or multilamellar spheroid structures”

√ Many thanks for your suggestion. We greatly appreciate your kind thoughts. We have corrected the mistake.

Lines 467-468: Typo “catan-ionic“; „antiangio-genic”

√ Many thanks for your suggestion. We greatly appreciate your kind thoughts. We have corrected the mistake.

Line 510: It may be relevant to note that cleavable inkers are needed for case (ii).

√ Many thanks for your suggestion. We greatly appreciate your kind thoughts. And the thought has been added to the manuscript as below, “Among these API-ILs types, novel cleavable linkers may be used for the second type, which is one of the future research directions.” Thank you very much!

Fig. 9: Figure and legend should be plotted on the same page.

√ Many thanks for your suggestion. We greatly appreciate your kind thoughts. We have taken note of this issue. Thank you very much! 

All section 3: The biological fate of IL used for parenteral application should be discussed.

√ Many thanks for your suggestion. We greatly appreciate your kind thoughts. We have showed the use of ionic liquids in transdermal drug delivery systems in the literature 100 (lines 526-537) and 110 (lines 574-593). In addition, we have added supplementary content on the application of ILs in transdermal drug delivery systems in line 492-496.

Section 3.3: The authors only discuss alkyl chain length, but did not comment on charge effects.

√ Many thanks for your suggestion. We greatly appreciate your kind thoughts. We have supplemented the content on charge effects, which is shown in lines 703-711. Thank you very much!

Lines 580-583: Dots should be used at the end of the sentences.

√ Many thanks for your suggestion. We greatly appreciate your kind thoughts. We have corrected the mistake.

Fig.10: Size is to small and plotting quality is insufficient to allow readability of structural formulas and labels. The effects on DNA and biofilms as shown in panel C and D are not justified by citations and explanations in the text. Typical antimicrobial substances that affect cell membrane integrity do not necessarily intercalate with DNA.

√ Many thanks for your suggestion. We greatly appreciate your kind thoughts. This view is quoted from reference 147, which reviews the possible antibacterial mechanism of ionic liquids as novel antimicrobial agents. We have supplemented the content about panel C and D, which is shown in lines 717-728.

Lines 644-646: In previous sentences, the stabilizing effect is discussed, while here the word “destabilization” is used. Is this a typo? Also, the correlation of fluorescence characteristics and tertiary structure may not be obvious to all readers based on the phrasing that is used.

√ Many thanks for your suggestion. We greatly appreciate your kind thoughts. We have corrected the mistake. In addition, we have added supplementary information on fluorescence characteristics and tertiary structure. Thank you very much!

Lines 673-681: It is not clear, why the pharmaceutical field and the biomedical field are separated in this listing. It is also not clear, while topics such as drug delivery systems APIs or antimicrobial agents are assigned to the biomedical, rather than the pharmaceutical field.

√ Many thanks for your suggestion. We greatly appreciate your kind thoughts. We have corrected the statement.

Lines 685-700: It may be inappropriate to handle all types of IL similarly when speaking about toxicity. Some differentiation may be helpful as toxicity is related to the specific chemical compositions rather than the general categorization of a substance as ionic liquid.

√ Many thanks for your suggestion. We greatly appreciate your kind thoughts. This paragraph is trying to convey is the current lack of sufficient assessment of the potential toxicity and safety of ionic liquids. It the end, we have pointed out that it is best to select non-toxic anions and cations with good biocompatibility to prepare safer and effective ionic liquids and conduct a comprehensive biotoxicity evaluation.

Yours sincerely

Prof. Yong-Gang Zhao

College of Biological and Environmental Engineering, Zhejiang Shuren University, Hangzhou, 310015, China

E-mail address: [email protected]

Reviewer 2 Report

Comments and Suggestions for Authors

Unfortunately, the authors did not finish the work. I stated in the general part of the review that it needed a serious restructuring, but that did not happen. The authors answered the questions individually and only corrected them as if my first major comment did not exist. A serious restructuring of work is needed. Most of the material errors were removed because in such a structurally disordered work we could not notice the smallest errors. I will most kindly ask the authors to read the paper once more and think about whether they can further improve the structure and reduce repetition. Please concentrate on the mistakes, we did not repeat the same things more than once. For example, with Span and Tween, I say write both the chemical and pharmacopoeial names. You in one case chemically, in the other case pharmacopeia. this inconsistency stings the reader's eyes. Please read the entire paper once more and try to structure it as well as possible, reducing terminological errors. Carriers of the medicinal substance have been distributed to you, I don't know how. If you did not find an example wording, please write it and say that this approach is still in the early stages of research. And please reduce the repetitions—especially the ones about the benefits of ionic liquids. I asked you to correct microemulsions so that they can be used for other routes of administration, for example oral... I don't see that you have corrected. I regret that I am now prolonging the process of publishing this work, which has its own qualities. However, I have the impression that you did not take the three major revisions too seriously. I'm not saying that we reviewers are always right, but you really had serious mistakes and corrected exactly where we emphasized, and I have the impression that you haven't read the paper. I believe that this is still a major revision, but since you have improved the quality of the work, let it be a minor revision. I will not write individual comments on purpose so you can read the whole paper.

Comments on the Quality of English Language

Minor editing of English language required

Author Response

Dear reviewer,

We are truly grateful to your critical comments and thoughtful suggestions. Based on these comments and suggestions, we have made careful modifications on the original manuscript. All changes made to the text are in red color. We hope the new manuscript will meet your magazine’s standard. Below you will find our point-by-point responses to your comments/questions:

Comments and Suggestions for Authors

Unfortunately, the authors did not finish the work. I stated in the general part of the review that it needed a serious restructuring, but that did not happen. The authors answered the questions individually and only corrected them as if my first major comment did not exist. A serious restructuring of work is needed. Most of the material errors were removed because in such a structurally disordered work we could not notice the smallest errors. I will most kindly ask the authors to read the paper once more and think about whether they can further improve the structure and reduce repetition. Please concentrate on the mistakes, we did not repeat the same things more than once. For example, with Span and Tween, I say write both the chemical and pharmacopoeial names. You in one case chemically, in the other case pharmacopeia. this inconsistency stings the reader's eyes. Please read the entire paper once more and try to structure it as well as possible, reducing terminological errors. Carriers of the medicinal substance have been distributed to you, I don't know how. If you did not find an example wording, please write it and say that this approach is still in the early stages of research. And please reduce the repetitions—especially the ones about the benefits of ionic liquids. I asked you to correct microemulsions so that they can be used for other routes of administration, for example oral... I don't see that you have corrected. I regret that I am now prolonging the process of publishing this work, which has its own qualities. However, I have the impression that you did not take the three major revisions too seriously. I'm not saying that we reviewers are always right, but you really had serious mistakes and corrected exactly where we emphasized, and I have the impression that you haven't read the paper. I believe that this is still a major revision, but since you have improved the quality of the work, let it be a minor revision. I will not write individual comments on purpose so you can read the whole paper.

√ Many thanks for your suggestion, and we greatly appreciate your kind suggestion. We are very sorry for our previous responses to your comments/questions. In this version, we have carefully reviewed the comments/questions and have revised the manuscript accordingly. Thank you very much!

The work is really interesting and significant. However, the structure of the paper is completely wrong, as well as there are many terminological errors. Indeed when I got to the beginning of chapter 3.2. I realized that the structure absolutely cannot be like that, because what is at the beginning of 3.2. you write the whole chapter 3.1. I really think a major revision is needed. To be more clear, serious work restructuring and logical connection.

√ Many thanks for your suggestion, and we greatly appreciate your kind suggestion. Based on your suggestion, we have reworked the content of the two parts “3.1 and 3.2”, the content of part “3.2” should be a small part of “3.1”. And so, the content of part “3.2” has been changed to “3.1.5”. Thank you very much!

  1. Consider switching the order of the first and second paragraphs in the introduction. Somehow, it is more logical for me to go for a general division of ionic liquids, a division by generations... and then a specific application in pharmacy, biomedicine, and in general which generation would that be? To be more precise, I would transfer that first paragraph about the application in pharmacy later after the general description, where you mention it again.

√ Many thanks for your suggestion. We greatly appreciate your kind thoughts. Based on your suggestion (a general division of ionic liquids, a general division of ionic liquids and then a specific application), we have reworked and adjusted the content of the introduction to make it more logical as below, “Ionic liquids (ILs), which are also called molten salts, are a class of bulky and asymmetric organic cations and organic or inorganic anionic compounds, with melting points generally below 100 °C [1–3]. Compared with traditional organic solvents, ionic liquids have many unique properties, and their physical and chemical properties can be regulated by adjusting cations or anions [4,5]. Different combinations of cations and anions can form a huge number of ILs, and there are various ways to categorize them. According to the different ionic structure, traditional ILs are mainly divided into substituting imidazole, pyridine, pyrrolidine, quaternary ammonium salt, pyrrolidinium, quinolinium, etc. Commonly used anions are tetrafluoroborate, hexafluorophosphate, chloride, bromine, nitrate, acetic acid, methyl sulfate and so on (Figure 1) [6–8].

ILs with different chemical structure and properties can be briefly divided into three generations according to the order and age of their discovery. The research of ionic liquids has experienced the development from the first generation to the third genera-tion. The first generation mainly used in electroplating field was combined basically di-alkylimidazolium and alkylpyridinium cations with metal halide anions. These ILs with special physical properties,such as high thermal stability, low melting point and broad liquidity, can be used to prepare functional solvents to instead of some organic solvents. Most first-generation ILs suffer from low biodegradability, high toxicity to the aquatic environment and high preparation costs [9,10]. The second generation is stable in water and air, and it is synthesized from cations (e.g., dialkylimidazolium, alkylpyridinium, ammonium, and phosphonium) and anions (e.g., tetrafluoroborate, and hexafluoro-phosphate). These ILs with unique chemical properties can be used to prepare functional materials. By adjusting and modifying anions and cations and their substituents, physical and chemical properties such as melting point, viscosity, thermal stability, hydrophilic-ity, solubility, toxicity and biodegradability can be customized [11,12]. The third gener-ation of ILs employs some natural sources of anions (such as amino acids, fatty acids, etc.) and cations (such as choline). In addition to good physical and chemical properties, the third generation ILs also has low toxicity and good biodegradability. With the emer-gence of the third generation of ILs, the research on the application of ILs in bio-medicine is increasing gradually [11,13,14].

ILs have the advantages of designability, green non-toxicity, high stability, high solubility, and specific biological activities. The number of articles involving ILs is im-pressive and is growing significantly rapidly, and some of them show that ionic liquid can promote the reaction process, increase the yield and reduce the environmental pollution [2,11,15]. They have been successfully used in green solvents, drug delivery, and drug synthesis and other fields show great application prospects [16,17]. Further-more, the high tunability and good solubilization provide a new strategy for the prob-lems of poor solubility, unstable crystal form, poor biological activity and low drug de-livery efficiency in the pharmaceutical field. It is interesting to note that the focus of many IL studies now evolves in the direction of life sciences and medicine which lead to bio-medical applications of ILs become one of the major research trends (Figure 2) [17-22]”.

 Thank you very much!

  1. Line 175, consider whether the term therapeutic agents is correct, perhaps rather a compound with biological activity.

√ Many thanks for your suggestion. We greatly appreciate your kind thoughts. We have corrected the mistake. The “therapeutic agents” has been changed to “compounds with biological activity”. Thank you very much!

  1. Part 2.3. not only chemical modifications are made to increase solubility/dissolution rate/permeability and therefore bioavailability. This is also done by the wording, please correct it. For example. solid dispersions, lipid formulations, complexation with cyclodextrins.

√ Many thanks for your suggestion. We greatly appreciate your kind thoughts. We have corrected the mistake. The “To solve the problem of poor drug solubility, organic solubilizer (such as ethanol, methanol, acetone, and dimethyl sulfoxide) have traditionally been used to formulate drug molecules with limited water solubility and permeability. However, this initiative raises the additional issue that organic solvents or co-solvents with acute toxicity can remain in the drug product, which is not permitted by regulatory agencies [6].” has been corrected to “Traditionally, a number of strategies have been used to solve the problem of poor drug solubility, such as organic solubilizer, solid dispersions, lipid formulations, complexation with cyclodextrins, etc. [6]”

  1. 267 and 268 line reference 53, please check the terminology modified, controlled, extended release.

√ Many thanks for your suggestion. We greatly appreciate your kind thoughts. We have corrected the mistake.

  1. In chapter 2 the whole, especially 2.3. and 2.4. try to give more examples and specific significant results that have been achieved. As you mentioned reference 67 (lines 351-355), just think ie. check that this reference and this segment have a place in chapter 2.3. or 2.4. you state that the dissolution rate and solubility are increased (which is chapter 2.3). Nowhere in this example have you listed the characteristics of the polymorphic form?

√ Many thanks for your suggestion. We greatly appreciate your kind thoughts. We have moved reference 67 to chapter 2.3.

  1. Row 358, I don't like the term therapeutic effect... It is true that drug delivery systems have a medicinal substance in 99% of cases, which will have a therapeutic effect, but sometimes they also play a role in a certain diagnostic procedure.

√ Many thanks for your suggestion. We greatly appreciate your kind thoughts. We have corrected the mistake. And the “Drug delivery is the process of getting drugs into the body for therapeutic effect through an appropriate technical route.” has been changed to “Most drug delivery systems are the process of getting drugs into the body for therapeutic effect through an appropriate technical route. Sometimes, drug delivery system also plays a role in a certain diagnostic procedure”. Thank you very much!

  1. Line 359 is by no means an expression of the four types of systems, this is the four ways of applying the medicine..... And this vector is even a little questionable... But these are not types of systems, but ways of application.

√ Many thanks for your suggestion. We greatly appreciate your kind thoughts. We are very sorry for the mistake, and the wrong information has been deleted. Thank you very much!

  1. Reference 68, consider the term drug deprivation

√ Many thanks for your suggestion. We greatly appreciate your kind thoughts. The term “reduce drug deprivation and loss” has been changed to “reduce drug loss”. Thank you very much!

  1. Read lines 365-375 it seems that no formulation approach other than ionic liquids has produced the desired results. It's not like that. Ionic liquids have their own positive effects, give examples, but do not rule out other approaches, they have their own formulations on the drug market that are specifically used and give excellent results in therapy.

√ Many thanks for your suggestion. We greatly appreciate your kind thoughts. We are very sorry for the misleading expression, and We have corrected this misleading expression. “For a long time, researchers have been investigating extended-release formulations that provide optimal therapeutic concentrations of drugs to solve the problems faced by traditional DDSs. Due to their unique properties, such as good chemical stability and biocompatibility, ionic liquids have been widely used in biomedical fields, especially in DDSs [71–74]. In recent years, naturally-derived anions (such as amino acids, organic acids and fatty acids) and cations (such as choline, amino acid ester, glycine betaine, and protein-derived cations) have been designed and prepared as ILs for applying in DDSs due to their low toxicity and good biocompatibility [75–77].” has been corrected to “For a long time, researchers have been investigating extended-release formulations that provide optimal therapeutic concentrations of drugs. Due to the chemical stability and biocompatibility of ILs, they have attracted much more attention from scholars and provided a feasible way to solve the problems faced by traditional DDSs [71–74]. In recent years, naturally-derived anions (such as amino acids, organic acids and fatty acids) and cations (such as choline, amino acid ester, glycine betaine, and protein-derived cations) have been designed and prepared as ILs for applying in DDSs [75–77]. Although this approach has certain effects and feasibility, it is still in the early stages of research.”

Thank you very much!

  1. Please state if there is any formulation with ionic liquids on the market or in the clinical research phase and what are the results, what is the clinical research stage (you will understand that my question under 9 makes sense). Don't be too superlative about your topic, but a little more realistic comparison.

√ Many thanks for your suggestion. We greatly appreciate your kind thoughts. We have not collected more information about ionic liquids on the market or in the clinical research phase. ILs-DDSs may be currently only in the early stages of research. And we have mentioned this in the manuscript as below, “Due to the chemical stability and biocompatibility of ILs, they have attracted much more attention from scholars and provided a feasible way to solve the problems faced by traditional DDSs [71–74]. In recent years, naturally-derived anions (such as amino acids, organic acids and fatty acids) and cations (such as choline, amino acid ester, glycine betaine, and protein-derived cations) have been designed and prepared as ILs for applying in DDSs [75–77]. Although this approach has certain effects and feasibility, it is still in the early stages of research.” Thank you very much!

  1. Lines 377-385 have a lot of repetition...

√ Many thanks for your suggestion. We greatly appreciate your kind thoughts. We have simplified this paragraph as below, “Surfactant ionic liquids (SAILs) are a class of ionic liquids carrying short alkyl chains, which have a structure similar to that of cationic surfactants due to the presence of a charged hydrophilic head group and a non-polar hydrophobic tail group. They have attracted a lot of interest from researchers around the world [78-80]. The remarkable properties of SAIL, including high thermal stability and biodegradability, make it potentially useful as a drug delivery system [81–83]. Today SAILs are used to design stable drug-carrying vehicles as surfactants or co-surfactants, including micelles, vesicles, microemulsion systems, microporous polymer and active pharmaceutical ingredient (Figure 6) [70,84–89].” Thank you very much!

  1. Line 413 chapter title micelles and you talk about nanoparticles. It is absolutely not the same. Maybe this comment of mine is harsh, but somehow the whole paper seems to me technologically written, with a weaker application in pharmacy and medicine. It seems to me, the first time I looked, that none of the authors are pharmacists, possibly doctors. That's why you simply have such terminological errors. You did not give too many applicable examples. Errors in the type of route of administration and type of therapeutic systems. Some sentences are taken from papers and those in the context where one sentence is taken have a completely different meaning. There is no strong connection, especially when we talk about wording.

√ Many thanks for your suggestion. We greatly appreciate your kind thoughts. Just as you said, I was working at Ningbo Municipal Center for Disease Control and Prevention before working at Zhejiang Shuren University. Our group has been engaged in the preparation of nanomaterials and their application in pharmaceutical and biomedical analysis.

  1. Rapid determination of memantine in human plasma by using nanoring carboxyl-functionalized paramagnetic molecularly imprinted polymer d-μ-SPE and UFLC-MS/MS, Drug Testing and Analysis, 7 (2014) 535-543.
  2. Fast determination of catecholamines in human plasma using carboxyl-functionalized magnetic-carbon nanotube molecularly imprinted polymer followed by liquid chromatography-tandem quadrupole mass spectrometry, Journal of Chromatography A, 1429 (2016) 86–96.
  3. Application of carbon nanosorbent for PRiME pass-through cleanup of 10 selected local anesthetic drugs in human plasma samples, Analytica Chimica Acta 960 (2017) 72-80.
  4. Enhanced cleanup efficiency hydroxy functionalized-magnetic graphene oxide and its comparison with magnetic carboxyl-graphene for PRiME pass-through cleanup of strychnine and brucine in human plasma samples, Analytica Chimica Acta 1020 (2018) 41-50.
  5. Application of petal-shaped ionic liquids modified covalent organic frameworks for one step cleanup and extraction of general anesthetics in human plasma samples, Talanta 210 (2020) 120652.
  6. Simultaneous determination of 12 illicit drugs in human plasma by the PRiME clover-shaped nano-titania functionalized covalent organic frameworks pass-through cleanup procedure followed by ultra-performance liquid chromatography-tandem mass spectrometry, Journal of Chromatography A 1671 (2022) 463022.

In our previous work (Talanta 210 (2020) 120652), ionic liquids modified nanomaterials, i.e., ionic liquids modified covalent organic frameworks, have been used as a novel sorbent for cleanup and extraction of general anesthetics in human plasma samples. However, for ionic liquids in pharmaceutical and biomedical applications, our group really does not have a corresponding work basis. We especially hope to make the connection of this article much clearer with your help. Thank you very much!

  1. References 98.99 Read a little, clarify, and expand. This sounds weird

√ Many thanks for your suggestion. We greatly appreciate your kind thoughts. We have corrected the wrong expression, and the contents have changed as below, “Two types of microemulsions, water-in-oil (W/O) and oil-in-water (O/W), are widely used to develop effective delivery systems for polar and nonpolar drugs [98,99]. Sarkar et al. reported the formulation of an ionic liquid containing Tween-based nonaqueous microemulsion with biologically acceptable components, which indicates that the mi-croemulsion droplets are spherical in nature as their size varies linearly with the in-creasing ionic liquid content. And the nonaqueous IL/O microemulsion system can be used as a drug delivery system for many sparingly water-soluble drug molecules [98].” Thank you very much!

  1. Are microemulsions used only for skin application? There is no single system for oral administration and i.v. application? First... There are also registered preparations, especially self-micro emulsifying systems.

√ Many thanks for your suggestion. We greatly appreciate your kind thoughts. We have removed inappropriate range restrictions about the application of microemulsion.

  1. Row 443 in microemulsions, reference 83, nanosystems for substance delivery. Please read everything again and we don't have such illogicalities.

√ Many thanks for your suggestion. We greatly appreciate your kind thoughts. We have corrected the mistake. And “potential nanocarriers for transdermal drug delivery” has been corrected to “potential drug carriers for substance delivery”. Thank you very much!

  1. What is the drop size of microemulsion systems?

√ Many thanks for your suggestion. We greatly appreciate your kind thoughts. The radius of the microemulsion droplet is usually 10-100 nm.

  1. Please add the pharmacopoeial and chemical names in brackets for Tween and Span, the journal is from the pharmaceutical group.

√ Many thanks for your suggestion. We greatly appreciate your kind thoughts. We have added the information about Tween and Span as below, “The surfactant phase contains polyoxyethylene sorbitan monooleate (Tween-80) and sorbitan mono laurate (Span-20), and ethanol (1:1:1) as co-surfactant.” Thank you very much!

  1. 3.1.3. vesicular systems have a wide range of systems, try to be more specific

√ Many thanks for your suggestion. We greatly appreciate your kind thoughts. We have added more information about vesicular systems as below, “Vesicles, the unilamellar or multilamellar spheroid structures composed of amphiphilic molecules in aqueous medium, are size-selective filters with a structure similar to the biological membranes and can be used as models for biological membranes and in drug delivery. Vesicles are also called liposomes. Generally, if the amphiphilic molecules are synthetic surfactants, the structure formed is called a vesicle. Whereas, if the amphiphilic molecules are the natural surfactant lecithins, the structure formed is called a liposome.” Thank you very much!

  1. Line 479, 480 this first sentence is unacceptable nanosystems and microemulsions.... see the division... Microemulsions have a smaller drop size than nanoemulsions. Everything is mixed, please be careful.

√ Many thanks for your suggestion. We are very sorry for the mistake. We greatly appreciate your kind thoughts. And “there are also other types of nanoparticles as drug carrier applied in the drug delivery systems” has been corrected to “there are also other types of drug carrier applied in the drug delivery systems”. Thank you very much!

  1. Order 497 extended-release. Who? What? Medicinal substances. Who is? We can't just extend the release. The terminology is not aligned in your work and in a previous question, I mentioned a problem with the term extended-release.

√ Many thanks for your suggestion. We greatly appreciate your kind thoughts. We have corrected the mistake. The “The drug association efficiency of the developed system was up to about 76% in the presence of [Cho][Phe], and its sustained release was up to 72 h.” has been corrected to “The drug association efficiency of the developed system was up to about 76% in the presence of [Cho][Phe], and its extended-release of rutin was up to 72 h.” Thank you very much!

  1. Where did the beginning of 3.2 come from? Well, you wrote about the problem of poor solubility and permeability in 3.1. about solubilization and now again. Not really.

√ Many thanks for your suggestion. We greatly appreciate your kind thoughts. To some extent, both 3.1 and 3.2 address the problem of poor solubility of drugs. Based on your suggestion, we have reworked the content of the two parts “3.1 and 3.2”, the content of part “3.2” should be a small part of “3.1”. And so, the content of part “3.2” has been changed to “3.1.5”. Thank you very much!

Yours sincerely

Prof. Yong-Gang Zhao

College of Biological and Environmental Engineering, Zhejiang Shuren University, Hangzhou, 310015, China

E-mail address: [email protected]

Reviewer 3 Report

Comments and Suggestions for Authors

I welcome suggestions accepted by the authors.

Author Response

I welcome suggestions accepted by the authors.

√ Many thanks for your suggestion, and we greatly appreciate your nice comments on our article.

Round 3

Reviewer 1 Report

Comments and Suggestions for Authors

The authors mentioned the compound "Huimenthol" in their revision. I could not find information on the existense of this chemical in Web of Science, so a check by the authors is recommended.

Comments on the Quality of English Language

I also suggest an extensive english proof-reading as the language, sentence structure, and grammar is not always perfectly optimal. Other than that, I have no further comments and suggest acceptance after minor revision.

Author Response

Comments and Suggestions for Authors:

The authors mentioned the compound "Huimenthol" in their revision. I could not find information on the existense of this chemical in Web of Science, so a check by the authors is recommended.

√ Many thanks for your suggestion. We greatly appreciate your kind thoughts. We are very sorry for this mistake, and “Huimenthol” has been corrected to “menthol”. Thanks very much!

Comments on the Quality of English Language:

I also suggest an extensive English proof-reading as the language, sentence structure, and grammar is not always perfectly optimal. Other than that, I have no further comments and suggest acceptance after minor revision.

√ Many thanks for your suggestion. We greatly appreciate your nice comments on our manuscript. According to your suggestion, we have polished our manuscript carefully and corrected the grammatical, styling, and typos found in our manuscript.

Yours sincerely

Prof. Yong-Gang Zhao

College of Biological and Environmental Engineering, Zhejiang Shuren University, Hangzhou, 310015, China

E-mail address: [email protected]

Reviewer 2 Report

Comments and Suggestions for Authors

Accept in present form

Author Response

Comments and Suggestions for Authors:

Accept in present form

√ Many thanks for your suggestion, and we greatly appreciate your nice comments on our article.

Yours sincerely

Prof. Yong-Gang Zhao

College of Biological and Environmental Engineering, Zhejiang Shuren University, Hangzhou, 310015, China

E-mail address: [email protected]